# Multistep Quasimetric Learning for Scalable Goal-conditioned Reinforcement Learning

**Bill Chunyuan Zheng**[1]    **Vivek Myers**[1]    **Benjamin Eysenbach**[2]    **Sergey Levine**[1]

[1]UC Berkeley    [2] Princeton University

## Abstract

Learning how to reach goals in an environment is a longstanding challenge in AI, yet reasoning over long horizons remains a challenge for modern methods. The key question is how to estimate the temporal distance between pairs of observations. While temporal difference methods leverage local updates to provide optimality guarantees, they often perform worse than Monte Carlo methods that perform global updates (e.g., with multi-step returns), which lack such guarantees. We show how these approaches can be integrated into a practical offline GCRL method that fits a quasimetric distance using a multistep Monte-Carlo return. We show our method outperforms existing offline GCRL methods on long-horizon simulated tasks with up to 4000 steps, even with visual observations. We also demonstrate that our method can enable stitching in the real-world robotic manipulation domain (Bridge setup). Our approach is the first end-to-end offline GCRL method that enables multistep stitching in this real-world manipulation domain from an unlabeled offline dataset of visual observations and demonstrate robust horizon generalization.[1]

## 1    Introduction

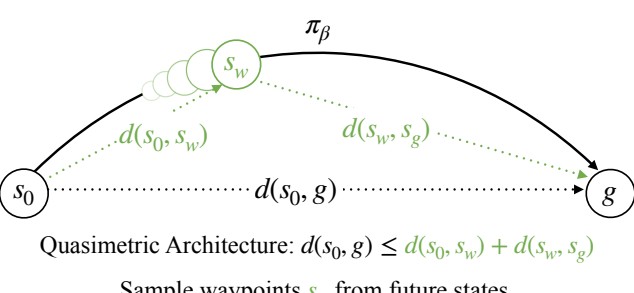

Quasimetric Architecture: $d(s_0, g) \leq d(s_0, s_w) + d(s_w, s_g)$

Sample waypoints $s_w$ from future states

Figure 1: In this paper, we present Multistep Quasimetric Estimation (MQE). Unlike prior work in quasimetric distance learning that use single-step TD updates (Wang et al., 2023) or contrastive learning-based Monte-Carlo updates (Myers et al., 2024), MQE is the *first* work to incorporate multistep returns with real-world success.

It is natural for humans to use inherent ideas of distances to represent task progress: a GPS will tell you how far you are from the destination and a cookbook will tell you how long a recipe will take. Humans can solve tasks by taking the shortest route possible (stitching) and combining several learned tasks together in sequence in a new environment (combinatorial generalization). The AI problem of reaching goals presents a rich structure (formally, an optimal substructure property) that can be exploited to decompose hard problems into easy problems, and reinforcement learning (RL) has been used to address such problems. However, many past attempts at leveraging this property in modern high-dimensional, stochastic settings still need much work. Current offline approaches tend to separate TD learning (local updates) (Mnih et al., 2013; Kostrikov et al., 2022; Kumar et al., 2020)

---

[1]Website and code: `https://mqe-paper.github.io`

and MC learning (global value propagation) (Eysenbach et al., 2022; Myers et al., 2024); as TD learning can theoretically recover the optimal Q function ($Q^*$), while MC methods can only recover the behavior Q function ($Q_\beta$), but tend to work well in practice.

However, the effectiveness of both the temporal difference and Monte Carlo methods degrades, often from a combination of increasing the horizon length for TD methods (Park et al., 2025b) or difficulties in finding the optimal temporal distance (Park et al., 2024a). These combined challenges present an intriguing opportunity to find methods that can leverage the advantages of both approaches, where one can perform both local and global value propagation at once.

Our main contribution lies in **M**ultistep **Q**uasimetric **E**stimation (MQE), an offline GCRL method that incorporates both multistep value learning and quasimetric architectures without needing explicit hierarchy. By leveraging such a unique combinations of the benefits of TD and MC methods, MQE allows the learned policy to (*i*): display a much stronger level of horizon generalization compared to previous methods, which allow us to demonstrate the desired "stitching" behavior, (*ii*): provide a stable method that can extract strong goal-reaching policies even from noisy data, and (*iii*): such stability in training allows it to be applied in real-world robot learning problems without additional design choices. To our knowledge, MQE is the first method capable of taking advantage of multistep TD returns with global value propagation through quasimetric architectures. MQE achieves SoTA performance on tasks that require complex control and long-horizon reasoning (up to 21 DoF and 4000 timesteps respectively), and in real world robotic settings, MQE displays compositional generalization behaviors that are not seen in previous RL algorithms.

## 2 RELATED WORK

Our work build upon previous works in offline RL and temporal distance learning.

**Goal-conditioned Reinforcement Learning.** Recent work has studied the stitching and horizon generalization capabilities of GCRL. Park et al. (2025b) show that while offline RL is easy to scale on short-horizon tasks, it is difficult to learn long-horizon tasks that require more complex reasoning within the agent, which can easily deviate from the optimal distance due to compounding TD errors. Other findings have shown that combinatorial generalization and stitching do not necessarily need dynamic programming (Ghugare et al., 2024; Brandfonbrener et al., 2023; Garg et al., 2023), which gives promise to using simpler methods for learning stitchable policies.

**Offline RL for Robotics.** We demonstrate offline GCRL scaling to real-world robotic tasks. Although reinforcement learning has been used to obtain highly capable specialist policies in various embodiments (Luo et al., 2025; Ball et al., 2023; Seo et al., 2025; Smith et al., 2023), behavior cloning still remains the most capable method for training generalist policies (O'Neill et al., 2024). Efforts have been made to allow self-supervised and offline RL in robotics (Zheng et al., 2024), however, directly training a policy with offline RL remains difficult, as many researchers have instead used other ways to incorporate RL, such as rejection sampling (Nakamoto et al., 2025; Wagenmaker et al., 2025) or data set curation (Mark et al., 2024; Xu et al., 2024). We show that our method can use multistep quasimetric learning to obtain effective policies for real-world robotic tasks, outperforming non-GCRL methods on long-horizon manipulation.

**Multistep RL.** RL using multistep returns has been widely used in online RL and offline-to-online RL settings (Li et al., 2025; Tian et al., 2025). This is desirable because in on-policy settings, performing RL with multistep return is theoretically correct and yields better performance (Munos et al., 2016; Asis et al., 2018; Schulman et al., 2016). However, in an offline setting, the theory behind such correctness breaks down (Watkins, 1989), as the learning paradigm becomes an inherently off-policy manner. Methods that use the entire trajectory in a Monte Carlo manner and attempt to recover the Q function can also be seen as a multistep RL algorithm (Eysenbach et al., 2022). However, such approach can only recover the behavior Q function $Q_\beta$ (Myers et al., 2024; Eysenbach et al., 2022).

**Temporal Distance Learning.** Our work is closely related by works that focus on successor representations (Dayan, 1993; Blier et al., 2021) and uses them to learn the probability of reaching goals. Previous works have shown that by using contrastive learning (Myers et al., 2024), one can

recover a temporal distance suitable for goal-reaching. Other works have also shown that using contrastive learning as a way to parameterize distance learning may also recover behavior Q function (Eysenbach et al., 2022). Previous works (Eysenbach et al., 2021; Myers et al., 2025c) have shown that it is possible to learn an optimal goal-reaching policy and Q function in theory. Our work differs in that we learn a Bellman optimal Q function under the bias of behavioral dynamics. This tradeoff helps us achieve considerable empirical gains in long-horizon goal-reaching tasks, and shows compositionality in real-world robotic learning problems.

## 3 PRELIMINARIES

In this section, we define temporal distances and our learning objective.

**Notation.** We consider a controlled Markov process (CMP) $\mathcal{M}$ with state space $\mathcal{S}$, action space $\mathcal{A}$, transition dynamics $\mathrm{P}(s' \mid s, a)$, and discount factor as $\gamma$. We consider goal-reaching policies $\pi(a \mid s, g) : \mathcal{S} \times \mathcal{S} \to \mathcal{A} \in \Pi$. We denote the behavior policy as $\pi_\beta$. In lieu of rewards, we optimize the maximum discounted likelihood of a policy reaching the goal, in which we can represent the goal-conditioned Q function and the value function as:

$$Q_g^\pi(s, a) \triangleq \mathbb{E}_{\{\mathfrak{s}_t, \mathfrak{a}_t\} \sim \pi} \Big[ \sum_{t=0}^\infty \gamma^t \, \mathrm{P}(\mathfrak{s}_t = g \mid \mathfrak{s}_0 = s, \mathfrak{a}_0 = a) \Big]. \tag{1}$$

$$V_g^\pi(s) \triangleq \mathbb{E}_{\{\mathfrak{s}_t\} \sim \pi} \Big[ \sum_{t=0}^\infty \gamma^t \, \mathrm{P}(\mathfrak{s}_t = g \mid \mathfrak{s}_0 = s) \Big]. \tag{2}$$

Equivalently, we can define the optimal Q-function as $Q_g^*(s, a) \triangleq \max_{\pi \in \Pi} Q_g^\pi(s, a)$. Previous work have shown that using contrastive leaning (Eysenbach et al., 2022; van den Oord et al., 2019) can recover the behavior distance, but not the optimal distance.

Similarly, prior work on MC learning (Myers et al., 2024; Eysenbach et al., 2022; 2021) has demonstrated that future states can be used as goals. We use geometric distribution as described in Eq. (3). This allows us to classify any future state in trajectory as goals, providing a robust way of learning goal-reaching policies.

$$\mathfrak{s}_t^+ \triangleq \mathfrak{s}_{t+K}, K \sim \mathrm{Geom}(1 - \gamma). \tag{3}$$

**Quasimetric distance representations.** Traditionally, offline RL algorithms use neural networks to represent the critic and value function $Q_g(s, a)$ and $V_g(s)$ (Kumar et al., 2020; Kostrikov et al., 2022). Separately, other works have been using dot products $Q(s, a, g) \triangleq \varphi(s, a)^\mathsf{T} \psi(g)$ (Eysenbach et al., 2022; Zheng et al., 2024) or geometric norms $\|\varphi(s) - \psi(g)\|_k$ for suitable values of $k$ (Eysenbach et al., 2024; Park et al., 2024c). We use quasimetric architectures (Wang et al., 2023; Pitis et al., 2020) to parameterize our value functions.

Formally, the space of quasimetric distances $\mathcal{Q}_\mathcal{X}$ over a set $\mathcal{X}$ is defined as the set of functions $d : \mathcal{X} \times \mathcal{X} \to \mathbb{R}_{\geq 0}$ that satisfy the following properties for all $x, y, z \in \mathcal{X}$ (Cobzaş, 2013):

(i) **Non-negativity:** $d(x, y) \geq 0$.
(ii) **Identity:** $d(x, x) = 0$.
(iii) **Triangle Inequality:** $d(x, z) \leq d(x, y) + d(y, z)$.

We will also use the notation $\mathcal{D}_\mathcal{X}$ to denote the more general class of distances that satisfy only (i) and (ii).

To enforce the quasimetric properties of the successor distances, we use the Metric Residual Network (MRN) architecture (Liu et al., 2023), which parameterizes a space of quasimetrics in terms of a learned representation, as shown in Eq. (4). MRN splits the representation into $N$ equally sized components, and in each part, takes the sum of an asymmetric component (maximum of ReLU) and a symmetric component ($l_2$ norm) of the difference between the two embeddings $x$ and $y$.

$$d_{\text{MRN}}(x,y) \triangleq \frac{1}{N}\sum_{k=1}^{N} \max_{m=1\dots M} \max(0, x_{kM+m} - y_{kM+m}) + \|x_{kM+m} - y_{kM+m}\|_2 \quad (4)$$

Given any function $f : \mathcal{X} \to \mathbb{R}^{NM}$, Eq. (4) defines a quasimetric distance on $\mathcal{X}$. Liu et al. (2023) show that for sufficiently large $M$, this parameterization is universal. Thus, fitting a quasimetric distance function can be accomplished by fitting $d_{\text{MRN}}(f(x), f(y))$ for a sufficiently expressive class of representations $f \in \Phi$. We make use of this architecture with a learned representation over $\mathcal{S} \times \mathcal{A} \cup \mathcal{S}$ to express our goal-conditioned Q function $Q_g(s,a)$ and value function $V_g(s)$ as distances between states/state-action pairs and goals, as demonstrated in (Myers et al., 2025c).

## 4 MULTISTEP QUASIMETRIC ESTIMATION (MQE)

In this section, we develop MQE framework based on quasimetric architectures, which can be broken down into 3 parts: (1) multistep backup under a quasimetric architecture, (2) imposing action invariance as a form of value learning, (3) practical implementation details for extracting the goal-reaching policy. To the best of our knowledge, MQE is the **first** method to effectively combine multistep returns with a quasimetric distance parameterization, and achieves superior results to previous methods that use TD returns, contrastive learning for value estimation, or hierarchical methods.

**Definitions.** We will define a distance (quasi)metric over states and state-action pairs that is proportional to the goal-conditioned $Q$ and $V$ functions (Myers et al., 2024):

$$Q_g(s,a) = V_g(g)e^{-d((s,a),g)}, \qquad V_g(s) = V_g(g)e^{-d(s,g)}. \quad (5)$$

We learn a parameterized form of this distance with learned state and state-action $\psi(s) \in \mathbb{R}^{NM}$ and $\phi(s,a) \in \mathbb{R}^{NM}$ representations and a quasimetric distance function (4): :

$$d\big((s,a),g\big) \triangleq d_{\text{MRN}}\big(\varphi(s,a), \psi(g)\big), \qquad d(s,g) \triangleq d_{\text{MRN}}\big(\psi(s), \psi(g)\big). \quad (6)$$

### 4.1 MULTISTEP RETURNS WITH QUASIMETRIC ARCHITECTURE

The first design principle of MQE is to use multistep returns under a quasimetric architecture. To that end, we start with the fitted onestep Q iteration that operates on quasimetric distance representations, in which $\leftarrow$ denotes regressing from the LHS to the RHS:

$$e^{-d((s,a),g)} \leftarrow \mathbb{E}_{\{s' \sim \text{P}(s'|s,a)\} \sim \mathcal{D}}\big[\gamma \cdot e^{-d(s',g)}\big]. \quad (7)$$

This is similar to optimizing the critic in traditional RL (Kostrikov et al., 2022; Lillicrap et al., 2019), the main difference being that we use a quasimetric architecture to represent the Q and value function. Due to the quasimetric architecture of the network, the reward signal is embedded within the parameterized distance, as $V_g(s)$ is defined to have a value of $V_g(g) > 0$ when $s = g$. We denote this regression objective as $\mathcal{T}$.

Our insight for the $\mathcal{T}$ objective described in Eq. (7) is that, instead of applying this invariance to only the future state $s' \sim \text{P}(s' \mid s,a)$, we can extend this principle to any state between the current state and goal. This transforms a onestep optimization into a on-policy multistep optimization procedure. To do so, we first define the shorthand $\mathfrak{s}_t^w$, which refers to a "**waypoint**" between the current state and goal state. Empirically, we find that using a combination of Bernoulli distribution and geometric distribution (described in Eq. (8)), capped at the index of the future state work the best.

$$s_t^w \leftarrow \mathfrak{s}_{t+k'} \quad \text{for} \quad \begin{cases} k' \sim \min(\text{Geom}(1-\lambda), K) & \text{with probability } 1-p, \\ k' = 1 & \text{with probability } p. \end{cases} \quad (8)$$

We now optimize the same objective as in Eq. (7), but across any such waypoint we sample. To account for the multistep nature of this objective, we modify Eq. (7) below to accommodate such changes.

$$e^{-d((s,a),g)} \leftarrow \mathbb{E}_{\{(s_t,a_t), s_t^w\} \sim \mathcal{D}}\big[\gamma^{k'} \cdot e^{-d(s_t^w,g)}\big]. \quad (9)$$

We denote this new objective as $\mathcal{T}_\beta$, as the future state of the sample is restricted by trajectories generated by $\pi_\beta$. This is similar to the $n$-step returns in prior work (Sutton, 1988; Munos et al., 2016; Li et al., 2025), although we do not sample future states with a fixed number of steps. In practice, we can use any loss function to make the LHS equal to the RHS (in expectation) concerning Eq. (7) and Eq. (9). We use a form of Bregman divergence with LINEX losses (Parsian and Kirmani, 2002), as it does not incur vanishing gradients when the two distances have become close in value (Banerjee et al., 2004; Myers et al., 2025c):

$$D_T(d, d') \triangleq \exp(d - d') - d' \tag{10}$$

Using this loss, we can concretely define both $\mathcal{L}_{\mathcal{T}_\beta}$ and $\mathcal{L}_{\mathcal{T}}$ that can optimize the objectives of $\mathcal{T}$ and $\mathcal{T}_\beta$.

$$\mathcal{L}_{\mathcal{T}_\beta}\left(\phi, \psi; \{s_i, a_i, s_i^w, g_i, k_i'\}_{i=1}^N\right) = \sum_{i=1}^N \sum_{j=1}^N D_T\big(d((s_i, a_i), g_j), d(s_i^w, g_j)\big) - k_i' \log \gamma). \tag{11}$$

These two losses, when applied to our quasimetric network, will allow us to propagate the value in either a onestep or multistep manner.

**Remark: relationship between $p$, multistep backup $\mathcal{T}_\beta$, and onestep backup $\mathcal{T}$.** By changing $p$, we can adjust the distance (in expectation) from the waypoint from the current state, as $p = 1$ means that $\mathcal{T}_\beta$ is equivalent to $\mathcal{T}$ (as $k' = 1$). We demonstrate in Section 5.3 why only using $\mathcal{L}_{\mathcal{T}}$ is insufficient for learning a good distance representation and goal-conditioned policy.

## 4.2 Learning Value Functions via Enforcing Action Invariance

With respect to quasimetric networks, (Myers et al., 2024) has demonstrated that if the training data is collected using a Markovian policy, then the optimal critic should observe the property of *Action Invariance*. As a result, the optimal critic should observe the following property (Sutton and Barto, 2018):

$$V_g^*(s) = \max_{a \in \mathcal{A}} Q_g^*(s, a). \tag{12}$$

This can be satisfied, under our construction, if $d(s, (s, a)) = 0$ for each action $a \in \mathcal{A}$ (Myers et al., 2025c) (action invariance). Since our construction of Q and value function do not observe this property, we need to enforce it using gradient descent, as described in Eq. (13).

$$d(\psi(s), \varphi(s, a)) \leftarrow 0 \tag{13}$$

We remark that this is also similar to the value loss seen in other offline RL methods that employ both a value and Q function, namely IQL (Kostrikov et al., 2022) and TMD (Myers et al., 2025c), and the desired outcome of Eq. (13) is similar to the value loss described in IQL when $\tau \approx 1$. One common failure case of basic regressions, such as using $\mathcal{L}_1$ or $\mathcal{L}_2$ regression for Eq. (13), is that the algorithm quickly converges to trivial solution in which $\varphi(s, a) = \psi(s) = 0$, which fails to learn any valuable distance measures. As a result, TMD requires a hyperparameter $\zeta$ to regulate the magnitude of gradients of the invariance loss. To counteract this, we use a smoother formulation that allows more relaxed enforcement of Eq. (13) when the violation is low in magnitude. As a result, we can define $\mathcal{L}_{\mathcal{I}}$ as:

$$\mathcal{L}_{\mathcal{I}}\big(\varphi, \psi; \{s_i, a_i\}_{i=1}^N\big) = \sum_{i,j=1}^N \left(e^{-d(\psi(s_i), \varphi(s_i, a_j))} - 1\right)^2. \tag{14}$$

Now the loss will scale with the magnitude of such deviation, which removes the need of hyperparameter tuning for an appropriate multiplier as well as stabilizing training dynamics.

### 4.3 POLICY EXTRACTION

We extract the goal-conditioned policy $\pi(s, g) : \mathcal{S}^2 \to \mathcal{A}$ using the learned distance using the behavior-regularized deep-deterministic policy gradient (DDPG + BC) (Fujimoto and Gu, 2021):

$$\mathcal{L}_\mu(\pi; \{s_i, a_i, g_i\}_{i=1}^N) = \mathbb{E}\Big[\sum_{i,j=1}^N d\big((s_i, \pi(s_i, g_j)), g_j\big) - \alpha \log \pi(a_i \mid s_i, g_i)\Big]. \tag{15}$$

Given that a smaller $d$ measure correspond to a higher Q value, as defined in Eq. (5), we can maximize the Q values by minimizing the distance produced by our quasimetric network. We tune the BC coefficient $\alpha$ per environment. We provide more hyperparameter details in Section D.

### 4.4 IMPLEMENTATION DETAILS & ALGORITHM

We concisely define our final learning objective in Algorithm 1. Unlike previous works in hierarchical RL (Nachum et al., 2018; Park et al., 2024b), we randomly sample these waypoints, and we learn a single critic $\mathcal{Q}$ that operates on $\mathcal{S}$ and $\mathcal{A}$ and a single goal-reaching policy $\pi_\mu$. As a result, our method does not contain any additional components and is simpler to implement than these hierarchical methods.

---

**Algorithm 1:** Multistep Quasimetric Estimation

---

**Require:** Dataset $\mathcal{D}$, Batch size $B$, training iteration $T$, Probability $p$

1: Initialize quasimetric network $\mathcal{Q}$ with parameters $(\varphi, \psi)$, goal-reaching policy $\pi_\mu$
2: **for** $t = 1...T$ **do**
3:     Sample $\{s_i, a_i, s_i', s_i^w, g_i\}_{i=1}^B \sim \mathcal{D}$                                        (8)
4:     Update $\mathcal{Q}$ with multistep backup by minimizing $\mathcal{L}_{\mathcal{T}_\beta}(\varphi, \psi; \{s_i, a_i, s_i^w, k_i'\}_{i=1}^B)$     (9)
5:     Update $\mathcal{Q}$ with action invariance constraints by minimizing $\mathcal{L}_{\mathcal{I}}(\varphi, \psi; \{s_i, a_i\}_{i=1}^B)$    (14)
6:     Update policy $\pi_\mu$ with DDPG+BC by minimizing $\mathcal{L}_\mu(\pi_\mu; \{s_i, a_i, g_i\}_{i=1}^B)$      (15)
7: **return** $\pi_\mu$

---

### 4.5 ANALYSIS

The key theoretical result we show is that this algorithm is able to perform policy improvement on top of the behavior policy $\pi_\beta$ in a tabular setting under standard assumptions. The tabular version of MQE can be viewed as executing three steps:

1. Minimize Eqs. (11) and (14) under the data distribution from $\pi_\beta$:

$$\min_{\hat{d} \in \mathcal{D}_{\mathcal{S} \times \mathcal{A} \cup \mathcal{S}}} \mathbb{E}_{(s,a) \sim p^{\pi_\beta}, g \sim p^{\pi_\beta}} \Big[ D_T\big(d((s,a), g), \mathbb{E}_{p^{\pi_\beta}(s^w|s,a)}[\gamma^{k'} e^{-d(s^w, g)}]\big) + (e^{-d(s, (s,a))} - 1)^2\Big]. \tag{16}$$

2. Constrain the distance to be a quasimetric. Mathematically, this can be expressed as projecting the distance into a quasimetric space via a *path relaxation operator* $\mathcal{P} : \mathcal{D}_{\mathcal{X}} \to \mathcal{Q}_{\mathcal{X}}$ (Myers et al., 2025a), defined as

$$\mathcal{P}(d)(x, z) \triangleq \min_{y \in \mathcal{X}}\big[d(x, y) + d(y, z)\big]. \tag{17}$$

Applying this to the learned distance, we construct $\tilde{d} = \mathcal{P}\hat{d}$.

3. Extract the policy via Eq. (15):

$$\min_\pi \mathbb{E}_{(s,a) \sim p^{\pi_\beta}, g \sim p^{\pi_\beta}} \Big[ \tilde{d}\big((s, \pi(s, g)), g\big)\Big]. \tag{18}$$

This statement is formalized in Theorem 1 below.

**Theorem 1.** *Suppose behavior policy $\pi_\beta$ induces full support over state action pairs $(s, a) \sim d^{\pi_\beta}$ in a tabular setting. We fit a distance by minimizing Eq. (16) and extract a policy by minimizing Eq. (18). Then the extracted policy $\pi$ satisfies $V_g^\pi(s) \geq V_g^{\pi_\beta}(s)$ for all $s, g \in \mathcal{S}$.*

The proof is provided in Section C.

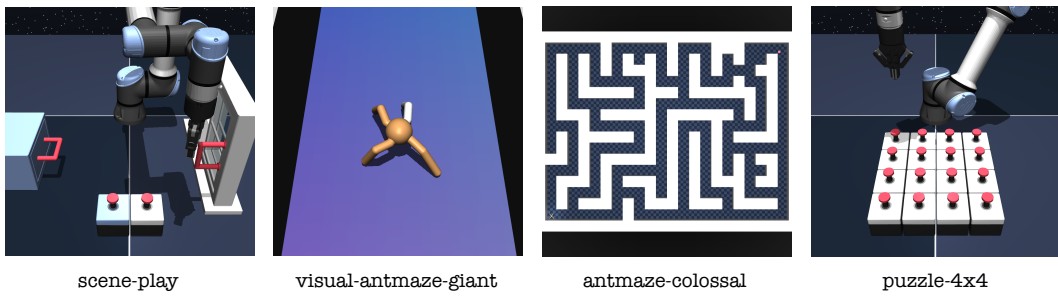

scene-play     visual-antmaze-giant     antmaze-colossal     puzzle-4x4

Figure 2: Tasks from various state and pixel-based environments for OGBench. `Antmaze-colossal` is 50% larger than any other mazes available on OGBench, and in `stitch` datasets, test the agent's ability to generalize over horizon that is up to **1000%** longer.

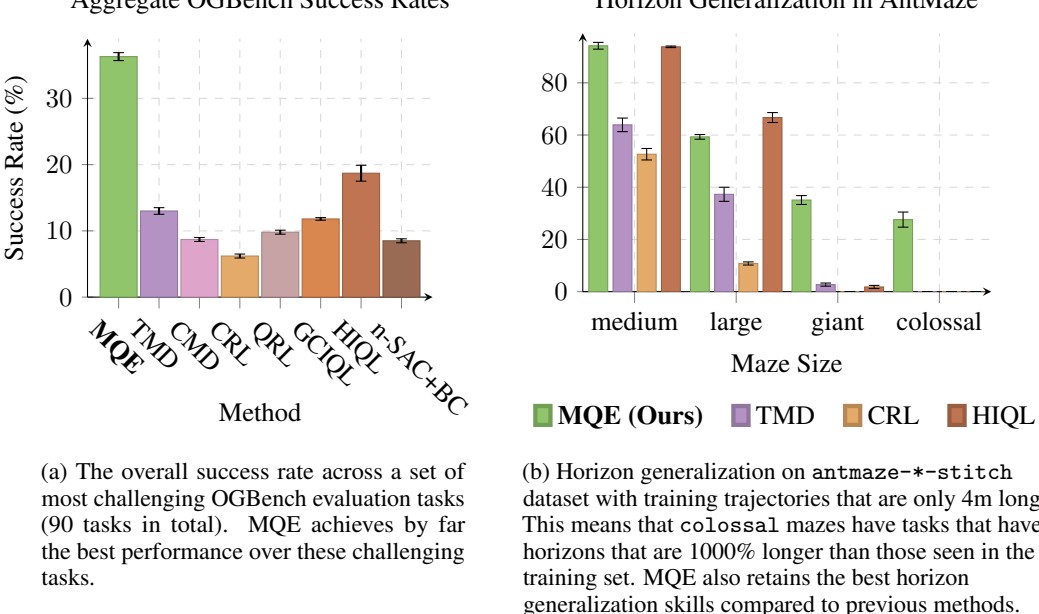

(a) The overall success rate across a set of most challenging OGBench evaluation tasks (90 tasks in total). MQE achieves by far the best performance over these challenging tasks.

(b) Horizon generalization on `antmaze-*-stitch` dataset with training trajectories that are only 4m long. This means that `colossal` mazes have tasks that have horizons that are 1000% longer than those seen in the training set. MQE also retains the best horizon generalization skills compared to previous methods.

Figure 3: Comparisons of MQE against prior methods on OGBench.

## 5 EXPERIMENTS

Our goal of experiments is to understand the benefits MQE brings when it comes to enabling a policy to generalize compositionally (execute multiple tasks seen in training separately together) and in terms of horizon (generalize over a longer task when a similar but shorter task was seen in training set). To that end, we pose the following questions:

1. How much does MQE improve the horizon generalization abilities of agents?
2. What qualitative improvements does MQE bring in terms of compositional generalization?
3. What are the design decisions to ensure the success of MQE?

**Experiment setup** Our experiments use challenging, long-horizon problems in offline RL benchmarks as well as real-world settings. We use OGBench (Park et al., 2025a) for our experiments on simulated benchmarks and the BridgeData setup (Walke et al., 2024) for our real-world evaluation.

### 5.1 SIMULATED EVALUATION ON OFFLINE GOAL-REACHING TASKS

We evaluate MQE in both locomotion and manipulation in OGBench (Park et al., 2025a). For locomotion tasks, in addition to the three standard sized mazes, we also designed a "colossal"-sized

maze. This maze is 50% larger than that of the "giant"-sized mazes currently available on OGBench, and it requires as many as 4000 steps for an agent to traverse through the entire maze (see Fig. 2). We employ 13 state-based environments and 5 pixel-based environments (each pixel-based environment takes in a $64 \times 64 \times 3$ observation) with 5 tasks each, bringing a total of **95** tasks to evaluate in our OGBench setup.

We compare against the following baselines: GCIQL (Kostrikov et al., 2022), CRL (Eysenbach et al., 2022), QRL (Wang et al., 2023), HIQL (Park et al., 2024b), nSAC+BC (Park et al., 2025b; Haarnoja et al., 2018), CMD (Myers et al., 2024), and TMD (Myers et al., 2025c). These methods use either only TD learning (GCIQL, QRL, nSAC+BC), MC value estimation via contrastive learning (CRL, CMD, TMD), use horizon reduction techniques (nSAC+BC with value horizon, HIQL with policy horizon) or use a quasimetric architecture for distance learning (QRL, CMD, TMD). We detail how these methods are implemented in Section D. By comparing against these methods, we can gain a better outlook on *what* advantage MQE has over other works that learn distances only locally, globally, or in a hierarchical manner.

Table 4 and Fig. 3 show the performance of MQE across state- and pixel-based environments on OGBench. In general, MQE exhibits considerably better capabilities of extracting goal-reaching policies, and in some instances (such as `humanoidmaze-giant-stitch`, exhibits a $10\times$ improvement over the previous best methods, including HIQL and n-SAC+BC, which does explicit policy horizon reduction. The only exception to this is vision-based manipulation, where HIQL performed better. We also compared the performance of MQE against TMD, CRL, and HIQL on `antmaze` of various sizes, with the training set fixed to only 4 meters long via the usage of `stitch` datasets. We show that as the evaluation horizon becomes longer, MQE still exhibits strong horizon generalization, and it is the *only* method that still exhibits nonzero success rates in `colossal` mazes.

## 5.2 Real-World Evaluation of MQE

While OGBench environments focus on learning long-horizon tasks using mixed quality data in a single environment, we can use real-world BridgeData tasks to evaluate a more sophisticated kind of compositionality. BridgeData tasks consist of individual object manipulation primitives (e.g. picking up a banana and placing it on a plate), and our evaluation tasks are significantly longer (up to 4x), requiring the composition of multiple tasks in the dataset (e.g. placing four different objects on the plate) (Walke et al., 2024). The dataset *does not contain any example trajectories* that compose multiple tasks in this way. Accomplishing this sort of temporal composition is an important goal in offline RL, because it allows "stitching" long behaviors out of shorter chunks.

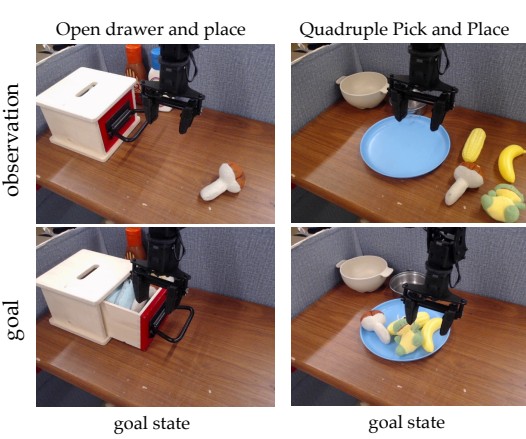

Figure 4: We evaluate MQE on multi-stage manipulation tasks on BridgeData. Below are examples of the starting observations and goal images.

We designed the tasks on BridgeData to test whether policies can compose multiple tasks at once without external guidance. Instead of instructing the policy to complete a single pick and place (abbreviated as PnP), we evaluate a policy's performance with PnP of up to 4 objects in sequence. To our knowledge, such a task in BridgeData **has never been completed without the use of hierarchical policies or high-level planners**. We also evaluate the policy on tasks requiring dependencies (i.e. the second task can only succeed when the first task succeeds), with the policy being tasked with opening a drawer and then placing the item within the drawer all conditioned by one image of an opened drawer with an item inside. Only **one previous work** (Myers et al., 2025b) has shown consistent success when using an end-to-end policy, noting the challenges involved with this type of policy. Examples of the tasks are shown in Fig. 4.

Table 1: Ablation results



(a) $\mathcal{L}_{\mathcal{I}}$ Ablation

| Configuration | Success Rate (%) |
|---|---|
| With $\mathcal{L}_{\mathcal{I}}$ | $\mathbf{26.5}^{(\pm 1.3)}$ |
| Without $\mathcal{L}_{\mathcal{I}}$ | $7.9^{(\pm 0.7)}$ |
| Using expectile $\kappa = 0.7$ | $11.3^{(\pm 1.1)}$ |
| Using expectile $\kappa = 0.9$ | $8.8^{(\pm 0.7)}$ |

(b) $\mathfrak{s}_t^w$ Sampling Ablation

| Configuration | Success Rate (%) |
|---|---|
| $k' \sim$ Eq. (8) | $\mathbf{26.5}^{(\pm 1.3)}$ |
| $k' \sim \text{Geom}(1 - \lambda)$ | $22.1^{(\pm 1.1)}$ |
| $k' \sim \text{Unif}[1, K]$ | $18.9^{(\pm 0.9)}$ |
| $k' \sim \text{Unif}[1, 50]$ | $17.8^{(\pm 1.3)}$ |
| $k' = 50$ | $1.7^{(\pm 0.5)}$ |
| $k' = 1$ | $0^{(\pm 0.0)}$ |



We use a 6DoF, 5Hz, WidowX250 manipulator for our robot learning tasks and we train and deploy a policy $\pi(a \mid s, g)$ conditioned on observations and goal images. We compare against the following methods: GCBC (Ding et al., 2020), GCIQL, and TRA (Myers et al., 2025b). We compare MQE against GCIQL, a commonly used offline RL method, and we compare MQE against both GCBC (Ding et al., 2020) and TRA (Myers et al., 2025b). TRA is designed for following both goal images and language instructions, but we only use goal images as the modality to test. We provide more details on policy training in Section E.1.

As in prior work that focused on long-horizon manipulation tasks (Black et al., 2024; Shi et al., 2025), we use task progress to measure the effectiveness of these policies due to the long-horizon nature of these tasks. We detail more on the experimental setup and how we assign progress in Section E.1.

Fig. 5 reports the *overall task progress* on 2 single-stage tasks and 4 tasks that require compositionality, and Table 6 reports the binary success rate of each task. We provide further analysis of policy rollout in Section E. Here, we observe that while MQE helps with single-stage tasks (single PnP, open drawer) against GCBC, both TRA and GCIQL can still perform competitively. However, as the number

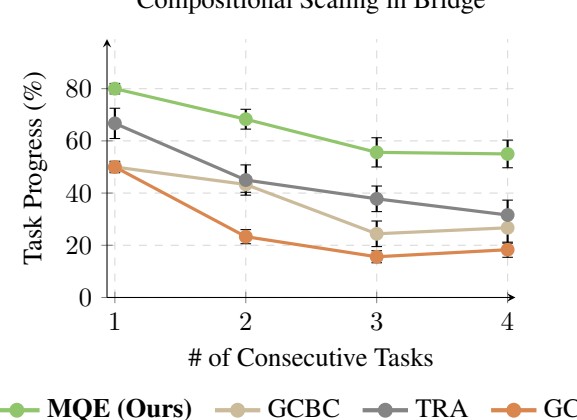

Figure 5: Task progress on BridgeData tasks based on how many consecutive tasks the agent is required to perform, plotted with both the mean and the standard error bars.

of tasks needed to be performed in sequence increases, we see that MQE is able to retain a relatively high task progress, while both GCIQL and TRA's performance regressed.

Taking a look at the two most difficult tasks, we have `quadruple PnP` and `drawer open and place`. These tasks are the most challenging since quadruple PnP required the agent to reason 4 consecutive primitives together, and drawer open and place required the agent to complete the first task (open the drawer) before completing the second task (putting the mushroom in the drawer). Among all methods that we have tested, only MQE and TRA displayed positive success rate, as demonstrated in Table 6. We provide more details on policy rollouts in Section E.

## 5.3 ABLATION STUDIES

In this section, we explore the design choices involved for MQE. To that end, we investigate the heuristics needed for MQE. We use the `humanoidmaze-giant-stitch` environment and dataset, and explore the following design questions:

- How do different distributions for sampling the waypoint $s_t^w$ affect MQE's success?
- Is the objective of action invariance $\mathcal{I}$ necessary in MQE?
- How do $\lambda$ and $p$, the two hyperparameters that affect the geometric sampling of $s_t^w$, change MQE's performance?

**Should we use action invariance for value learning?** Table 1a shows that when using the same set of hyperparameters, imposing action invariance as an explicit term helps to learn a much better critic, as compared to only using $\mathcal{L}_{\mathcal{T}_\beta}$. Additionally, we also implemented value learning via expectile loss, described in (Kostrikov et al., 2022). While expectile loss performed well as a value learning objective, it performed considerably worse than action invariance, which validates the theoretical results shown in Section C.

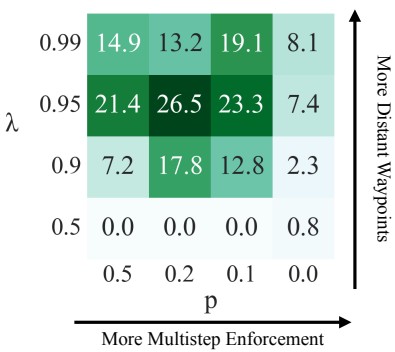

Figure 6: Sucess rate of MQE on `humanoidmaze_giant_stitch` using $\alpha = 0.01$, averaged over 4 seeds.

**How should we sample our waypoints?** We first evaluate the best distribution to sample future waypoints. Table 1b shows that when using a geometric distribution, we achieved much better performance than using a uniform distribution or using a fixed interval. We believe that since the goals are sampled via a geometric distribution, matching the waypoint with another geometric distribution helps the network to generalize better between the two separate observations.

We then investigate how to combine the hyperparameters $\lambda$ and $p$ for best performance. Fig. 6 provides an illustration of success rate over pairs of $(p, \lambda)$. The figure suggests that both hyperparameters need to be relatively high in value. This indicates that MQE needs: (1) a waypoint far enough for the value to rapidly propagate and (2) a high enough $p$ to ensure that local consistencies are being respected. We also see that if we increase the value of the sampling coefficient $p$, MQE cannot learn a good policy for goal-reaching. This shows that $\mathcal{T}$ is *not* sufficient to learn a good distance for policy learning, as the agent did not learn a good distance representation.

## 6 CONCLUSION

We introduced Multistep Quasimetric Estimation (MQE), a novel method that combines the benefits of fast value propagation via multistep backup and the global constraint of quasimetric distances. MQE is able to solve challenging and long-horizon tasks in simulated benchmarks and on a real-world robotic manipulator.

**Limitations and Future Work.** While MQE achieves strong performance, we sample the waypoints based on heuristics. This could incur more computation costs when finding the optimal way of sampling the waypoint for environments that are outside of our evaluation range. Future work can investigate the theoretical connection between sampling waypoints and successor distances, investigate the effect of such policy learning on different policy classes such as action-chunking policies, and apply the same method across methods beyond offline RL in scenarios such as offline-to-online RL or online RL.

**Acknowledgements** We would like to thank Seohong Park, Qiyang Li, Michael Psenka, and Kwanyoung Park for helpful discussions. This material is based upon work supported by the National Science Foundation under Award No. 2441665 and ONR N00014-25-1-2060. Any opinions, findings and conclusions or recommendations expressed in this material are those of the author(s) and do not necessarily reflect the views of the NSF and ONR.

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

## A    WEBSITE AND CODE

We provide the website, alongside the full implementation of MQE, TMD, CMD, and the new antmaze-colossal mazes on OGBench at the following URL: `https://mqe-paper.github.io`.

## B    LLM USAGE STATEMENT

We used large language models for proofreading and correcting grammatical errors, as well as providing suggestions for formatting visualizations. Any ideas generated in the paper are all entirely created by the authors and no LLMs are involved in that aspect.

## C    PROOF OF POLICY IMPROVEMENT

Here, we provide the full proof of Theorem 1.

**Theorem 1.** *Suppose behavior policy $\pi_\beta$ induces full support over state action pairs $(s, a) \sim d^{\pi_\beta}$ in a tabular setting. We fit a distance by minimizing Eq. (16) and extract a policy by minimizing Eq. (18). Then the extracted policy $\pi$ satisfies $V_g^\pi(s) \geq V_g^{\pi_\beta}(s)$ for all $s, g \in \mathcal{S}$.*

*Proof of Theorem 1.* We can express the algorithm described in Section 4.5 as the following two steps (combining the quasimetric projection and Eq. 18):

$$(1) \quad \min_{\hat{d} \in \mathcal{D}_{\mathcal{S} \times \mathcal{A} \cup \mathcal{S}}} \mathbb{E}_{(s,a) \sim p^{\pi_\beta}, g \sim p^{\pi_\beta}} \left[ D_T\big(d((s,a), g), \mathbb{E}_{p^{\pi_\beta}(s^w | s, a)}[\gamma^{k'} e^{-d(s^w, g)}]\big) + (e^{-d(s, (s,a))} - 1)^2 \right]$$

$$(2) \quad \min_{\pi} \ \mathbb{E}_{(s,a) \sim p^{\pi_\beta}, g \sim p^{\pi_\beta}} \left[ \mathcal{P}\hat{d}\big((s, \pi(s, g)), g\big) \right]$$

Define $d_{\text{SD}}^{\pi_\beta}$ to be the *modified successor distance* induced by the behavior policy $\pi_\beta$, as defined by Myers et al. (2025c, Eq. (7)).

The key is that the two terms in step (1) act on different parts of $\hat{d}$'s domain. The LHS term acts on $\mathcal{S} \times \mathcal{A} \cup \mathcal{S}$, while the RHS acts on $\mathcal{S} \times \mathcal{S} \cup \mathcal{A}$. The LHS regresses $\hat{d}$ towards $d_{\text{SD}}^{\pi_\beta}$ on $\mathcal{S} \times \mathcal{A} \cup \mathcal{S}$, while the RHS enforces action-invariance on $\mathcal{S} \times \mathcal{S} \cup \mathcal{A}$. The result is that $\hat{d} \leq d_{\text{SD}}^{\pi_\beta}$ while corresponding to the temporal distances of some family of goal-parameterized policies. It follows that since $\mathcal{P}$ is monotone decreasing, we have $\mathcal{P}\hat{d} \leq d_{\text{SD}}^{\pi_\beta}$, and this still corresponds to the temporal distances of valid goal-reaching policies as a consequence of the existence of a stationary optimal goal-reaching policy in MDPs/CMPs.

Step (2) therefore necessarily recovers a policy $\pi$ such that $d_{\text{SD}}^{\pi} \leq d_{\text{SD}}^{\pi_\beta}$. In other words, the extracted policy $\pi$ satisfies $V_g^{\pi}(s) \geq V_g^{\pi_\beta}(s)$ for all $s, g \in \mathcal{S}$.

$\square$

# D OGBENCH EXPERIMENT DETAILS

This section provides additional experiment details for the OGBench experiments in Section 5.1.

## D.1 BASELINE DETAILS

Here, we briefly describe the inner workings of each baseline on OGBench.

Goal-conditioned Implicit Q-learning (**GCIQL**) uses expectile regression to learn a value function. Quasimetric RL (**QRL**) learns a quasimetric distance using bootstrapping under a quasimetric architecture, and constrains the distance with one-step cost in *deterministic* settings. Contrastive RL (**CRL**) uses binary cross entropy to regress the critic (defined as a dot product) towards goals that are future states and repel those that are not. Contrastive Metric Distillation (**CMD**) uses InfoNCE loss (van den Oord et al., 2019) to recover $Q_\beta$. Temporal Metric Distillation (**TMD**) (Myers et al., 2025c) use contrastive learning to learn the behavior Q-function, and then tightens the bound to optimize towards $Q^*$. CMD and TMD enforce the quasimetric property of successor distance architecturally.

Additionally, we compare MQE against two horizon reduction methods, n-step Goal-Conditioned Soft Actor-Critic with Behavior Cloning (**n-SAC+BC**) and Hierarchical Implicit Q learning (**HIQL**), which explicitly reduce the policy and value horizon separately. n-SAC+BC is similar to SAC+BC, but with $n$-step updates as described in Eq. (19) for a sampled batch $\mathcal{B}$.

$$\mathcal{L}_Q = \mathbb{E}_{\{s_i, a_i, \ldots, s_{i+n}, a_{i+n}, g\} \sim \mathcal{B}}[\mathcal{L}_{\text{BCE}}(Q(s_h, a_h, g), \sum_{i=0}^{n-1} \gamma^i r(s_{h+i}, g) + \gamma^n \bar{Q}(s_{h+n}, \pi(s_{h+n}, g), g))]$$

$$(19)$$

Where $\mathcal{L}_{\text{BCE}}(x, y) \triangleq -y \log x - (1 - y) \log(1 - x)$, as it demonstrated superior performance in (Park et al., 2025b). **HIQL** trains the same value function as GCIQL, but the agent extracts a hierarchical policy using AWR-like objectives. All policies trained on OGBench are designed to return a multimodal Gaussian distribution $\mathcal{N}(\mu; \Sigma)$, and during inference time, the neural network produces the distribution, and the policy samples from that as action.

## D.2 TASK VISUALIZATION

We provide the tasks used for `antmaze-colossal` environment on Fig. 7. The maze itself is 24 blocks in height and 18 blocks in width, which is 50% larger than the `giant` mazes in each dimension.

## D.3 IMPLEMENTATION DETAILS & HYPERPARAMETERS

Table 2 details the common hyperparameters for all methods on OGBench. Table 3 shows the $\alpha$ regularization hyperparameter that was found to be the best for performance.

To regulate $\mathcal{L}_{\mathcal{I}}$, we impose a hyperparameter $\zeta$ to act as a multiplier. This ensures that across both visual and state-based environment, $\mathcal{L}_{\mathcal{I}}$ does not immediately overpower $\mathcal{L}_{\mathcal{T}_\beta}$. The final critic loss becomes:

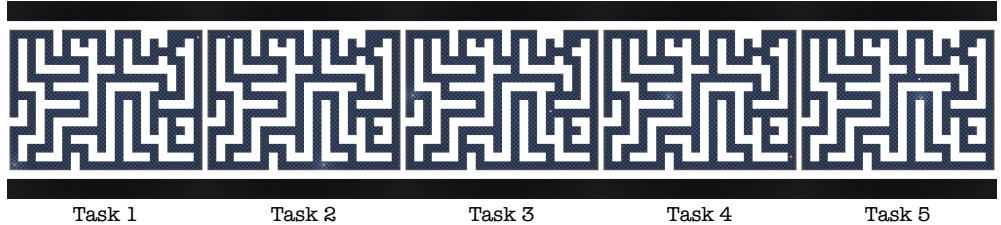

Figure 7: Task visualizations from `antmaze-colossal` environment. The ant occupies the starting position, and it must reach the red dot to complete the task.

Table 2: Network configuration for MQE on OGBench.

| Configuration | Value |
|---|---|
| batch size | 256 |
| latent dimension size | 512 |
| encoder MLP dimensions | $(512, 512, 512)$ |
| policy MLP dimensions | $(512, 512, 512)$ |
| layer norm in encoder MLPs | True |
| visual encoder (`visual-` envs) | `impala-small` |
| MRN components | 8 |
| Discount ($\gamma$) | 0.995 |
| Waypoint discount ($\lambda$) | 0.95 |
| Probability $p$ for sampling next state $s'_t$ for waypoint | 0.2 (state-based), 0.1 (pixel-based) |

Table 3: BC coefficient $\alpha$ for each environment

| Environment | $\alpha$ |
|---|---|
| `antmaze-navigate` | 0.1 |
| `antmaze-stitch` | 0.03 |
| `antmaze-explore` | 0.003 |
| `humanoidmaze` | 0.01 |
| `pointmaze` | 0.03 |
| `{cube,puzzle,scene-play}` | 1.0 |
| `{visual-{cube,puzzle,scene-play}}` | 3.0 |
| `*-noisy` | 0.1 |
| `visual-antmaze` | 0.3 |

Table 4: OGBench Evaluation

| | Methods | | | | | | | |
|---|---|---|---|---|---|---|---|---|
| Dataset | **MQE** | TMD | CMD | CRL | QRL | GCIQL | HIQL | n-SAC+BC |
| pointmaze_giant_navigate | $72.8^{(\pm2.5)}$ | $39.9^{(\pm5.2)}$ | $45.3^{(\pm3.7)}$ | $27.4^{(\pm3.4)}$ | $68.5^{(\pm2.8)}$ | $0.0^{(\pm0.0)}$ | $45.9^{(\pm3.0)}$ | $0.0^{(\pm0.0)}$ |
| pointmaze_giant_stitch | $59.2^{(\pm3.2)}$ | $9.1^{(\pm1.0)}$ | $8.1^{(\pm0.6)}$ | $0.0^{(\pm0.0)}$ | $49.7^{(\pm2.3)}$ | $0.0^{(\pm0.0)}$ | $0.0^{(\pm0.0)}$ | $0.4^{(\pm0.1)}$ |
| antmaze_large_explore | $67.7^{(\pm2.8)}$ | $0.9^{(\pm0.2)}$ | $0.8^{(\pm0.3)}$ | $0.3^{(\pm0.1)}$ | $0.0^{(\pm0.0)}$ | $0.4^{(\pm0.1)}$ | $3.9^{(\pm1.8)}$ | $0.2^{(\pm0.1)}$ |
| antmaze_giant_stitch | $35.1^{(\pm1.7)}$ | $2.7^{(\pm0.6)}$ | $2.0^{(\pm0.5)}$ | $0.0^{(\pm0.0)}$ | $0.4^{(\pm0.2)}$ | $0.0^{(\pm0.0)}$ | $1.8^{(\pm0.6)}$ | $9.2^{(\pm2.1)}$ |
| antmaze_colossal_navigate | $48.6^{(\pm2.4)}$ | $22.3^{(\pm1.1)}$ | $22.5^{(\pm3.1)}$ | $14.6^{(\pm1.8)}$ | $0.0^{(\pm0.0)}$ | $0.0^{(\pm0.0)}$ | $0.0^{(\pm0.0)}$ | $0.3^{(\pm0.1)}$ |
| antmaze_colossal_stitch | $27.6^{(\pm2.9)}$ | $0.0^{(\pm0.0)}$ | $0.2^{(\pm0.1)}$ | $0.0^{(\pm0.0)}$ | $0.0^{(\pm0.0)}$ | $0.0^{(\pm0.0)}$ | $0.0^{(\pm0.0)}$ | $0.5^{(\pm0.3)}$ |
| humanoidmaze_giant_navigate | $46.5^{(\pm2.5)}$ | $9.2^{(\pm1.1)}$ | $5.0^{(\pm0.8)}$ | $0.7^{(\pm0.1)}$ | $0.4^{(\pm0.1)}$ | $0.5^{(\pm0.1)}$ | $12.5^{(\pm1.5)}$ | $3.2^{(\pm0.5)}$ |
| humanoidmaze_giant_stitch | $26.5^{(\pm1.3)}$ | $6.3^{(\pm0.6)}$ | $0.2^{(\pm0.1)}$ | $1.5^{(\pm0.5)}$ | $0.4^{(\pm0.1)}$ | $1.5^{(\pm0.1)}$ | $3.3^{(\pm0.7)}$ | $1.7^{(\pm0.1)}$ |
| cube_double_play | $40.8^{(\pm1.2)}$ | $13.1^{(\pm2.3)}$ | $0.2^{(\pm0.1)}$ | $1.5^{(\pm0.5)}$ | $0.4^{(\pm0.1)}$ | $40.2^{(\pm1.7)}$ | $6.4^{(\pm0.7)}$ | $19.1^{(\pm0.3)}$ |
| cube_triple_noisy | $18.3^{(\pm2.2)}$ | $2.1^{(\pm0.6)}$ | $1.5^{(\pm0.5)}$ | $2.7^{(\pm0.5)}$ | $3.4^{(\pm0.4)}$ | $1.8^{(\pm0.2)}$ | $2.6^{(\pm0.4)}$ | $1.4^{(\pm0.9)}$ |
| puzzle_4x4_play | $18.7^{(\pm2.3)}$ | $10.0^{(\pm1.4)}$ | $0.2^{(\pm0.1)}$ | $1.5^{(\pm0.5)}$ | $0.4^{(\pm0.1)}$ | $25.7^{(\pm1.1)}$ | $7.4^{(\pm0.7)}$ | $11.4^{(\pm0.9)}$ |
| scene_play | $76.8^{(\pm2.1)}$ | $21.9^{(\pm1.9)}$ | $1.2^{(\pm0.4)}$ | $18.6^{(\pm0.8)}$ | $5.4^{(\pm0.3)}$ | $51.3^{(\pm1.5)}$ | $38.2^{(\pm0.9)}$ | $17.6^{(\pm1.4)}$ |
| scene_noisy | $30.8^{(\pm1.6)}$ | $19.6^{(\pm1.7)}$ | $4.0^{(\pm0.7)}$ | $1.2^{(\pm0.3)}$ | $9.1^{(\pm0.7)}$ | $25.9^{(\pm0.8)}$ | $25.2^{(\pm1.3)}$ | $19.1^{(\pm2.2)}$ |
| visual_scene_play | $38.1^{(\pm3.2)}$ | $20.7^{(\pm2.5)}$ | $16.1^{(\pm2.2)}$ | $9.6^{(\pm0.6)}$ | $5.4^{(\pm0.3)}$ | $12.2^{(\pm0.8)}$ | $49.9^{(\pm0.6)}$ | $7.1^{(\pm1.2)}$ |
| visual_cube_triple_play | $19.8^{(\pm0.9)}$ | $17.9^{(\pm1.3)}$ | $18.9^{(\pm1.1)}$ | $16.9^{(\pm1.1)}$ | $16.3^{(\pm0.3)}$ | $15.2^{(\pm0.6)}$ | $21.0^{(\pm0.2)}$ | $21.1^{(\pm2.4)}$ |
| visual_cube_double_noisy | $25.9^{(\pm1.6)}$ | $14.2^{(\pm1.3)}$ | $0.3^{(\pm0.3)}$ | $6.0^{(\pm1.4)}$ | $6.1^{(\pm1.2)}$ | $21.6^{(\pm0.9)}$ | $59.4^{(\pm1.6)}$ | $22.7^{(\pm1.1)}$ |
| visual_cube_triple_noisy | $25.0^{(\pm1.2)}$ | $17.7^{(\pm0.7)}$ | $16.1^{(\pm0.7)}$ | $15.6^{(\pm0.6)}$ | $8.6^{(\pm2.1)}$ | $12.5^{(\pm0.6)}$ | $21.0^{(\pm0.7)}$ | $17.1^{(\pm0.3)}$ |
| visual_puzzle_4x4_play | $17.9^{(\pm1.6)}$ | $9.8^{(\pm3.6)}$ | $7.2^{(\pm0.4)}$ | $9.6^{(\pm3.2)}$ | $0.0^{(\pm0.0)}$ | $16.2^{(\pm2.2)}$ | $60.1^{(\pm20.4)}$ | $10.3^{(\pm2.6)}$ |
| visual_antmaze_giant_stitch | $26.9^{(\pm3.1)}$ | $14.5^{(\pm2.5)}$ | $22.3^{(\pm1.9)}$ | $0.1^{(\pm0.1)}$ | $0.0^{(\pm0.0)}$ | $0.0^{(\pm0.0)}$ | $0.2^{(\pm0.1)}$ | $7.6^{(\pm1.1)}$ |
| Overall | $36.3^{(\pm0.6)}$ | $13.0^{(\pm0.5)}$ | $8.7^{(\pm0.3)}$ | $6.2^{(\pm0.3)}$ | $9.8^{(\pm0.3)}$ | $11.8^{(\pm0.2)}$ | $18.7^{(\pm1.2)}$ | $8.5^{(\pm0.3)}$ |

We **bold** the best performance. Success rate (%) is presented with the standard error across eight seeds for state-based environments and four seeds for pixel-based environments. All datasets contain 5 separate tasks each. We record the aggregate across all 5 tasks.

$$\mathcal{L}_{\mathcal{Q}}(\varphi, \psi; \{s_i, a_i, s_i^w\}_{i=1}^B) = \mathcal{L}_{\mathcal{T}_\beta}(\varphi, \psi; \{s_i, a_i, s_i^w, k_i'\}_{i=1}^B) + \mathcal{L}_{\mathcal{I}}(\varphi, \psi; \{s_i, a_i\}_{i=1}^B) \tag{20}$$

We use LINEX losses (Garg et al., 2023; Parsian and Kirmani, 2002) to parameterize the Bregman divergence in $\mathcal{L}_{\mathcal{T}_\beta}$, and to prevent exploding gradients, we set the

We use a batch size of 256 for MQE, and we use NVIDIA A6000 GPUs for our experiments. For state-based environments, it takes around 2.5 hours to finish both training and evaluation. For pixel-based environments, it takes 6 hours to complete both training and evaluation. During policy training, we found out that instead of using a larger batch size and randomly shuffle future states, as done by previous implementations (Park et al., 2025a), we achieved better performance by calculating the pairwise distance between each state and future state, at the cost of lower batch size but more comparisons between states.

## D.4  POLICY EVALUATION

We maintain an evaluation procedure similar to that of OGBench's. We evaluated each policy 50 times for the last 3 training epochs (800k, 900k, 1M step for state-based environments, 300k, 400k, 500k in pixel-based environments). For each of these evaluation epochs, use 8 seeds for every state-based environment and 4 seeds for every pixel-based environment.

## D.5  FULL RESULT TABLE

We record the full success rate of all tasks in OGBench in Table 4.

## D.6  VISUALIZATIONS

We provide visualizations on antmaze-large-explore to show the learned distances $d(s, g)$ for each $s$ in the environment with Fig. 8. We calculate the distance in IQL using the learned value function $V_g(s)$ directly. For CRL, we use the following equation to calculate the distance to match the BCE formulation being used in it: $d(s, g) = -\log \sigma(\varphi(s, 0)^\intercal \psi(g))$, where 0 corresponds to zero vector with the same dimension as the action. Given that CMD only learns one representation $\varphi(s, a)$, we parameterize the distance as $d = d_{\text{MRN}}(\varphi(s, 0), \varphi(g, 0))$. In TMD, we use $d(s, g) = d(\psi(s), \psi(g))$.

Out of all the heatmaps, we see that both GCIQL and CMD carry features that are similar to $L_2$ distance, while CRL and QRL contain many visual artifacts. GCIQL also has the problem of value propagation, where only a small region around the goal actually has a low distance measure. TMD has the best visualization out of all baselines, but still suffer from minor artifacts. In contrast, MQE recovers a distance that has the desired structure, which makes it desirable for learning a good goal-reaching policy.

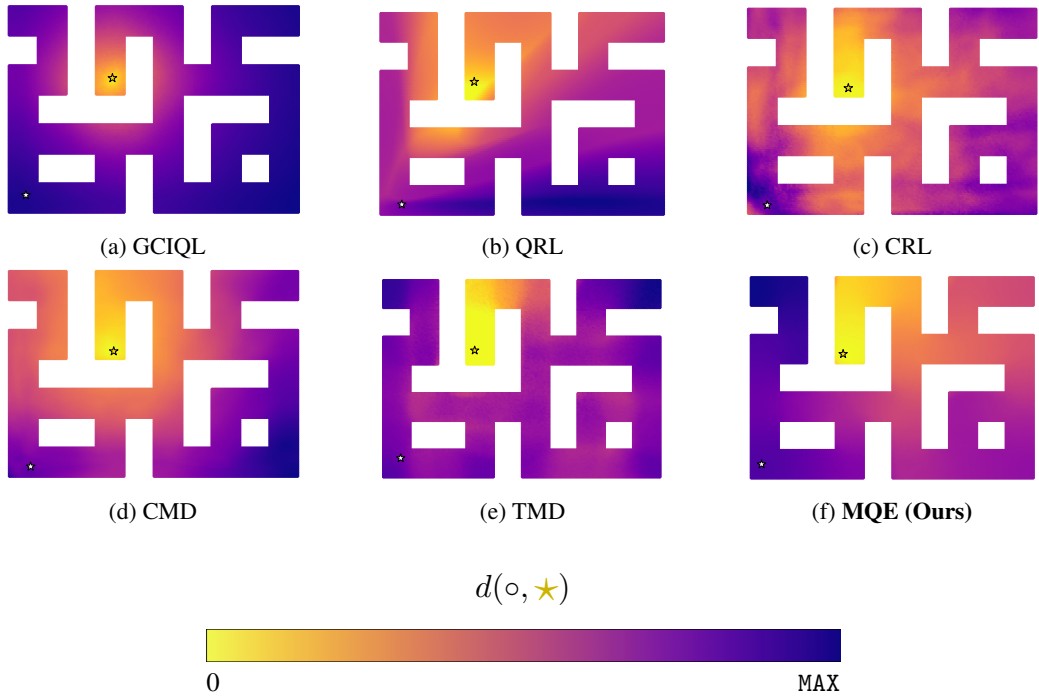

$$d(\circ, \star)$$

0          MAX

Figure 8: Comparisons of different learned distances. Brighter colors correspond to a lower distance measure. The agent starts at the position of the white star and the goal is set at the gold star. We compare especially the *structure* of the distance, as we normalize the ddpg loss in policy learning.

## E    BRIDGEDATA EXPERIMENT DETAILS

We evaluated each policy with a total of 15 trials each for each task. For each of the pick-and-place tasks, we assign 1 points for each successful pick and place (i.e. move a desired object to the desired location). Therefore, a policy can earn a maximum of $i$ points for each PnP task that manipulates $i$ objects.

For open drawer and place, we assign one point if the policy opens the drawer to the extent where the mushroom can be placed, but does not pull the drawer off the base completely, and assign another point if the policy is able to put the mushroom in the drawer.

### E.1    TRAINING CONFIGURATIONS

We use a pretrained ResNet34 as the backbone of the policy and encoders. We do not share the actor and critic encoder for both GCIQL and MQE. We co-train the actor and the critic for a total of 500,000 total steps with a batch size of 128, which takes around 60 hours when the model is trained on 4 v4-8 TPUs. For TRA, we produce the embedding of two separate encoders $\phi, \psi$, and align each other using symmetric InfoNCE loss. During inference, we use FiLM (Perez et al., 2017) to embed the learned goal representation into the actor, staying consistent with the implementation from (Myers et al., 2025b). In addition, we also describe the common hyperparameters used for BridgeData setup at Table 5.

Table 5: Network configuration for MQE on BridgeData.

| Configuration | Value |
|---|---|
| latent dimension size | 256 |
| encoder MLP dimensions | $(256, 256, 256)$ |
| policy MLP dimensions | $(256, 256, 256)$ |
| layer norm in encoder MLPs | True |
| MRN components | 8 |
| Discount $(\gamma)$ | 0.98 |
| Waypoint discount $(\lambda)$ | 0.95 |
| $p$ | 0.2 |

### E.2 TASK SUCCESSES

Table 6 records the success rate of each task across all six tasks that we evaluate, and Fig. 9 records the task progress for each task. In Table 6, we only measure whether each task has been completed to its fullest. While it does give out a stronger signal on whether a task displays nonzero success rate, it does not provide as much information on how the task was progressing overall.

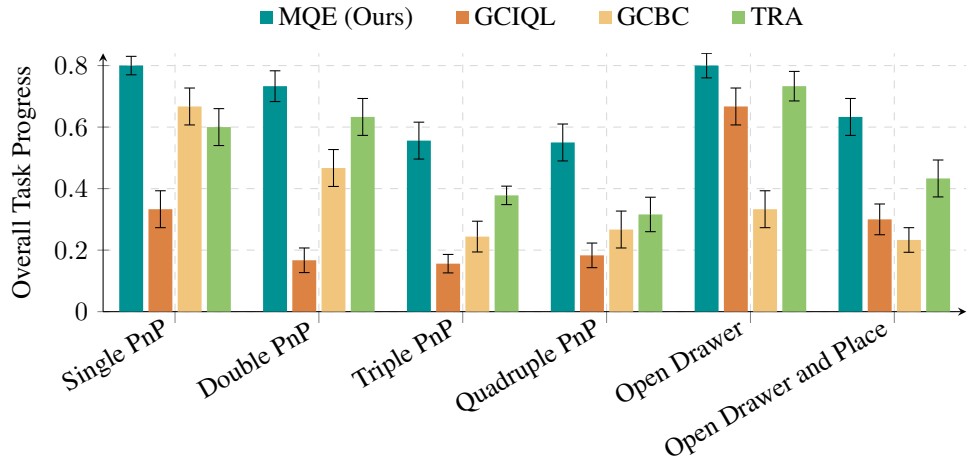

Figure 9: Task progress on BridgeData tasks; plotted with both the mean and the standard error bars.

Table 6: Binary success counts for each task.

| Task | MQE (Ours) | GCBC | GCIQL | TRA |
|---|---|---|---|---|
| Single PnP | 12/15 | 5/15 | 10/15 | 9/15 |
| Double PnP | 10/15 | 1/15 | 4/15 | 6/15 |
| Triple PnP | 4/15 | 0/15 | 0/15 | 1/15 |
| Quadruple PnP | 2/15 | 0/15 | 0/15 | 0/15 |
| Open Drawer | 12/15 | 10/15 | 5/15 | 11/15 |
| Open Drawer & Place | 5/15 | 0/15 | 0/15 | 1/15 |

## F POLICY ROLLOUT IN BRIDGEDATA

We provide the rollout of `triple PnP` in Fig. 10. We especially consider TRA because it also exhibits compositional generalization, yet there is no explicit policy improvement as compared to offline RL methods such as MQE.

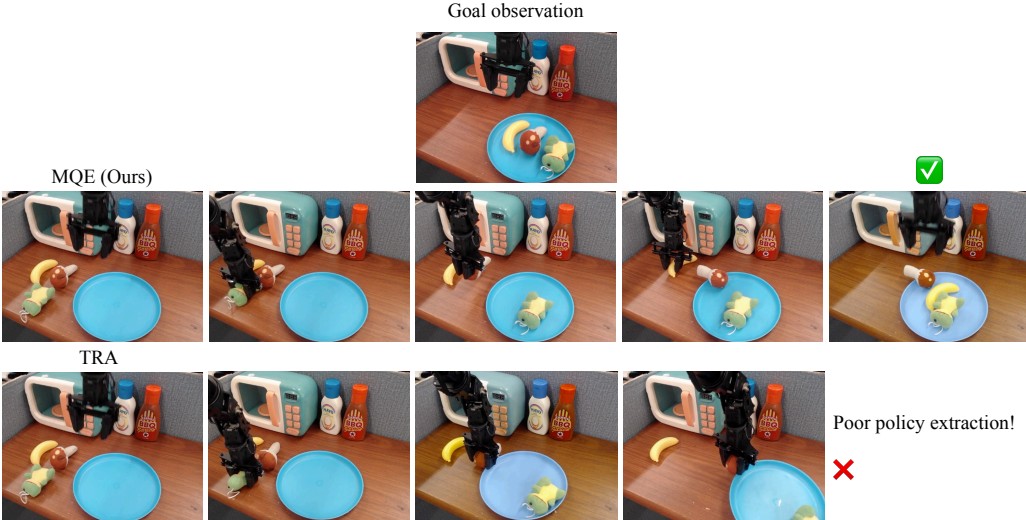

Figure 10: Inference from `triple pnp` task. We note that due to poor policy extraction and generalization, TRA is not able to complete the task.

