# OpenReview forum: "Scaling Goal-conditioned Reinforcement Learning with Multistep Quasimetric Distances"
_ICLR.cc/2026/Conference — ICLR 2026 Poster_

### Official Review · Reviewer_jD1e · 2025-10-26

**Soundness:** 3
**Presentation:** 2
**Contribution:** 1
**Rating:** 2
**Confidence:** 3

**Summary:**

This paper focuses on offline goal-reaching reinforcement learning, and pinpoints two families of methods: on-policy algorithms based on Monte-Carlo returns (and in several cases contrastive learning), and off-policy algorithms which enforce the known invariances of goal-conditioned value function through quasi-metric network architectures. This works extends TMD (Myers 2025), which previously united this two frameworks, by considering multi-step returns. Interestingly, this induces significant gains in performance across standard goal-conditioned tasks from the OGBench suite. The authors further perform an evaluation on the BRIDGE hardware setup, which confirms the strong performance of the algorithm. These results are accompanied by relevant ablations on the multi-step loss.

**Strengths:**

- Empirical results are very convincing, both in breadth (involving a large number of OGBench tasks as well as hardware experiments) and in outcomes (displaying large improvements over strong baselines).

**Weaknesses:**

- The proposed algorithm is strongly aligned with TMD, but in my opinion does not fully acknowledge this. To the best of my knowledge, each component in Section 4 (except for (11), which is a direct multi-step extension of the TD loss (12)) was already introduced in TMD. If I understand correctly, the final algorithm is simply TMD with a multi-step loss. If this is the case, it should be acknowledged in full.
- This works claims to combine off-policy and on-policy learning, but does not comment on whether the final value estimates are recovering the optimal value, or the value of the behavioral policy. Can the authors provide a formal discussion of what the proposed objectives recover, and why it is motivated?
- Related works are overall short (e.g. there is a single reference in the paragraph on GCRL), poorly formatted (several undefined references) and poorly written (the structure of the sentences in the last two paragraph is incorrect).

**Questions:**

### Minor issues and questions
- Line 12: remove "in"
- There are several broken or wrongly formatted references through the paper
- Line 122: this sentence is hard to follow: "X have demonstrated that Y instead of Z"
- Equation 7: the notation is unclear, as $s$ and $a$ appear under the expectation, as well as on the left side. formulating the expectation over $s'$ alone would be cleaner in my opinion.
- Equation 10: this choice seems to follow from TMD, which should be referred to in this case
- Figure 1: the caption refers to a 10x larger horizon, is this an overstatement? What is the length of optimal trajectories in giant and colossal mazes?
- What is the reasoning behind the selection of environments in Table 1? e.g. why is antmaze-large-explore evaluated instead of antmaze-large-navigate? The current selection appears somewhat arbitrary.
- Two of the ablation studies (matching the first two questions in 5.3) were relegated to the Appendix. This should be noted in the text.
- Table 3: how is the regularization parameter $\alpha$ tuned in each environment/algorithm combination?

### Conclusion
Despite the impressive empirical results, I currently lean towards rejection. To the best of my understanding, this method is an n-step extension of TMD, which is a minor contribution but is not problematic per se. My main concern is that this strong connection between the algorithms is not directly highlighted in the paper, which as a result seems to over claim its contribution. In am happy to further discuss whether my understanding is correct during the rebuttal phase. Moreover, unlike TMD, this works lacks an analysis of the objective and its solutions.

---

> ### Author Response · Authors · 2025-11-21
> **Response**
>
> Thank you for the thoughtful feedback. It seems your main concerns relate to novelty, presentation issues, and additional details on the experimental implementation. **We have revised the paper** based on the writing suggestions, **[added theoretical analysis](https://res.cloudinary.com/dp7qzzmt2/image/upload/v1763727124/MQE_Rebuttal-1763726716266_c8fjq1.png)** of the MQE objective, and **addressed concerns surrounding novelty**. We hope these revisions fully address your concerns.
>
> > The proposed algorithm is strongly aligned with TMD, but in my opinion does not fully acknowledge this. To the best of my knowledge, each component in Section 4 (except for (11), which is a direct multi-step extension of the TD loss (12)) was already introduced in TMD. If I understand correctly, the final algorithm is simply TMD with a multi-step loss. If this is the case, it should be acknowledged in full.
>
> We have added additional discussion in Section 4 showing the difference between TMD and MQE. MQE is designed to address key limitations of TMD: its inability to solve tasks like antmaze-giant-stitch (see Table 4 on the full results breakdown). We have revised the paper to more clearly delineate the contributions relative to TMD, which explains why MQE improves performance by 35% (on an absolute scale) on the most challenging tasks.
>
> The core difference between MQE and TMD is that the $\mathcal{T}$ backup used in TMD is only for enforcing consistency between the state-action pair and the next state (which is a part of contraction on $Q\_\\beta$), whereas we perform value (distance) propagation via multistep returns. As a result, we found that MQE does not require a contrastive component at all and can use the backup objective to learn the distances. We also modified the action invariance objective such that it no longer requires an extra hyperparameter $\\zeta$ as a loss multiplier. We want to note that this invariance objective is not only applicable to TMD, but this is also seen in other methods such as the value loss in IQL when $\\tau \\approx 1$ [1].
>
> We have also included [Theorem 1](https://res.cloudinary.com/dp7qzzmt2/image/upload/v1763727124/MQE\_Rebuttal-1763726716266\_c8fjq1.png) in our manuscript in Section 4, which shows that MQE performs policy improvement upon $\\pi\_\\beta$. While TMD can recover $Q^*$ in theory, in practice it performs at a considerably worse level compared to MQE. We used notation similar to TMD because MQE and TMD both fit quasimetric architectures over state and state-action pairs, and thus make use of a similar mapping of standard RL conventions (e.g., $Q$, $V$) to the equivalent notions over distances.
>
> Additionally, we have implemented a new variation of TMD based on your description, where we used a multistep backup instead of the $\\mathcal{T}$ backup. The results are shown in the table below for some of the state-based environments, with a substantial decrease in performance compared to the original implementation of TMD. A few experiments have not yet finished, hence the missing entries. We will continue to update the table throughout the rebuttal process, although the experiments conducted so far indicate that a multistep loss on TMD does not improve the performance of the policy.
>
>
> | Dataset                          | TMD       | TMD w/ Multistep Backup Loss       | MQE |
> | ------------------------------------ | ------------ | ---------------------------------- | ------------ |
> | `pointmaze\_giant\_navigate`     | 39.9 (±5.2) | 8.1 (±1.2) | 72.8 (±2.5) |
> | `pointmaze\_giant\_stitch`       | 9.1 (±0.0) | 0.0 (±0.0) | 59.2 (±3.2) |
> | `antmaze\_large\_explore`        | 0.9 (±0.2) | 0.0 (±0.0)  | 67.7 (±2.8)  |
> | `antmaze\_giant\_stitch`        | 2.7 (±0.6) | 0.0 (±0.0) | 35.1 (±1.7)  |
> | `antmaze\_colossal\_navigate`          | 22.3 (±1.1) | 8.4 (±0.9) | 48.6 (±2.4) |
> | `humanoidmaze\_giant\_navigate`      | 9.2 (±1.1)  | 4.4 (±1.2) | 46.5 (±2.5)  |
> | `cube\_double\_play`      | 13.1 (±2.3)  | 2.6 (±1.1) | 40.8 (±1.2)  |
> | `scene\_noisy` | 19.6 (±1.7)  | 22.1 (±1.7) | 30.8 (±1.6)  |
>
>
> > This works claims to combine off-policy and on-policy learning, but does not comment on whether the final value estimates are recovering the optimal value, or the value of the behavioral policy. Can the authors provide a formal discussion of what the proposed objectives recover, and why it is motivated?
>
> We have included a theoretical justification ([Theorem 1](https://res.cloudinary.com/dp7qzzmt2/image/upload/v1763727124/MQE\_Rebuttal-1763726716266\_c8fjq1.png)) in our manuscript (Section 4.5 in our methodology and proof in Appendix B), in which we note that MQE will recover a Q function that is better than the behavior Q function $Q\_\\beta$.

---

> ### Author Response · Authors · 2025-11-21
> **Response (cont.)**
>
> > Figure 1: the caption refers to a 10x larger horizon, is this an overstatement? What is the length of optimal trajectories in giant and colossal mazes?
>
> We would like to clarify that the `colossal` mazes are not 10x larger compared to previous mazes (they are 2.25x bigger than the `giant` mazes), but that the evaluation tasks in `colossal` mazes can be 10x longer than the ones seen in the `stitch` training datasets. The `stitch` datasets in locomotion environments in OGBench are designed to be 4 blocks long [2], and the 10x longer horizon compares against that statement. Based on our task definitions, tasks 1, 2, and 4 have an optimal path that is more than 40 blocks long, which means that the evaluation horizon is 10x longer than the seen horizon. We have reflected this change in Figure 3b of our revised manuscript.
>
> > What is the reasoning behind the selection of environments in Table 1? e.g. why is antmaze-large-explore evaluated instead of antmaze-large-navigate? The current selection appears somewhat arbitrary.
>
> We selected the environments based on the most challenging tasks for the available methods on OGBench with appropriate coverage of different baselines in all tasks (state and pixel-based locomotion and manipulation). In locomotion environments, this corresponds to the giant and colossal sized mazes for `navigate` and `stitch` datasets, and the `antmaze-large-explore` dataset (since it is the most difficult `explore` dataset available on OGBench). For manipulation tasks, we used a combination of `play` and `noisy` datasets across `cube, puzzle, scene` that are also challenging for all methods.
>
> > Table 3: how is the regularization parameter $\\alpha$ tuned in each environment/algorithm combination?
>
> We started off with the alpha value reported by CRL, and then chose 4-5 $\\alpha$ values that are near the value used by CRL, separated by multiples of 3 (i.e. if CRL used 0.03 as the alpha value, we sweep across $\\{0.003, 0.01, 0.03, 0.1, 0.3\\}$ as the potential hyperparameters). We then used two seeds per alpha value in our hyperparameter tuning procedure, before finding the best alpha value and then running it using 8 random seeds for evaluation. We emphasize that this is the same practice used by other offline RL methods [3,4], and we do not incur *additional computational costs*.
>
> This practice (tuning $\\alpha$ per environment over $\\{ 10^{-i}, 3\\cdot 10^{-i} \\}\_{i=0,1,\\dots}$) is used for all of the baselines implemented in the OGBench paper [2] and is in general *required* for offline RL algorithms to work in this setting by imposing conservatism toward $\\pi\_{\\beta}$. Future work should examine eliminating this parameter, but we believe the general problem in offline RL with policy extraction is beyond the scope of this work.
>
> > Equation 7: the notation is unclear, as $s$ and $a$ appear under the expectation, as well as on the left side. formulating the expectation over $s'$ alone would be cleaner in my opinion.
>
> Thank you. We have changed the equation to more clearly show the transition dynamics.
>
> > Equation 10: this choice seems to follow from TMD, which should be referred to in this case
>
> Thank you for pointing this out. We have modified our manuscript to appropriately attribute this.
>
> **Do these changes, together with the additional revisions and clarifications discussed below, fully address the reviewer's feedback about the paper?** If not, we look forward to continuing the discussion!
>
> ---
>
> [1] Kostrikov, I. et al., 2022. ''Offline Reinforcement Learning With Implicit Q Learning.'' *ICLR*
>
> [2] Park, S. et al., 2025. ''OGBench: Benchmarking Offline Goal-Conditoned RL.'' *ICLR*
>
> [3] Park, S. et al., 2025. ''Flow Q-Learning.'' *ICML*
>
> [4] Li, Q. et al., 2025. ''Reinforcement Learning With Action Chunking.'' *NeurIPS*

---

> ### Comment · Reviewer_jD1e · 2025-11-22
>
> Thank you for your response and for clarifying my questions on experiment design.
> The added theoretical justification is, in particular, very helpful.
> The paper has improved significantly in the revision, and I am happy to partially raise my score.
>
> I still lean towards rejection as the differences with respect to TMD are not entirely clear to me. As far as I can tell, there is three:
> - multi-step backups
> - modified action invariance objective
> - missing contrastive loss component
>
> Is my understanding correct? If so, why does the performance of TMD with multi-step backups degrade?
> When TMD is mentioned in the related works, I would encourage the authors to clearly state what are the differences, such that readers that are familiar with the literature can easily understand how this method is placed.
>
> As a side note, I find it confusing that only major changes in the revision are marked in color. It makes it hard to backtrack edits. Finally, the new related works section has incomplete sentences.

---

> ### Author Response · Authors · 2025-12-03
> **Additional Comparisons and Conceptual Differences**
>
> Thank you for the comment. We would like to more clearly show the differences between MQE and TMD in this response.
>
> You are correct in the sense that the three *main* differences between MQE and TMD, but there is also a key difference conceptually: while TMD tells us how we can recover the optimal successor distance *given a behavior distance* [1], MQE learns a new distance measure that is better than the behavior distance directly. As a result, TMD has better theoretical guarantees, but MQE is more stable empirically and leads to better performance. The TMD algorithm requires a learned behavior distance $d_{\text{SD}}^{\pi_\beta}$ (or equivalently $Q_\beta$) as input, and what the algorithm does is that it iteratively optimizes the behavior distance using the $\mathcal{T}$ and $\mathcal{I}$ operators to get $d_{\text{SD}}^*$ (we constrain the quasimetric property architecturally, which is also done in other works such as QRL and CMD). In practice, we often need to incorporate all of these elements into one training loop, in which contrastive learning is also included as a necessary component because there is no learned behavior distance.
>
> Our insight is that since the $\mathcal{T}$ component of TMD is essentially doing $Q_g(s, a) \leftarrow \mathbb{E}[r + \gamma V_g(s’)]$, we realized that $\mathcal{T}$ can directly be used to learn the requisite Q/value function parameterization via multistep backup. This changes the purpose of the operation entirely. We also note that the action invariance objective can also be interpreted as a value learning procedure based on the learned Q function, on top of the optimality guarantees that it has when applied in the TMD setting. We would like to reiterate that both of these operations have been used in normal, non-quasimetric RL settings (such as TD-n [2] and IQL) under different settings, and our main finding is that when these design choices are combined in such a way under quasimetric architecture, we can enable better horizon generalization abilities in simulation **and in real-world**.
>
> > Why does the performance of TMD with multi-step backups degrade?
>
> Since the backup component of TMD is a part that *optimizes* the learned behavior distance, by changing the single-step backup to the multistep backup, we lose the optimality guarantee. However, we also realized that we might have misinterpreted your statement, and we also ran experiments where we initialized the behavior distance using multistep backup. We present the results below:
>
> | Dataset                          | TMD       | TMD w/ Multistep Backup for $C(\pi)$      | MQE	|
> |----------------------------------|-------------|-------------|-------------|
> | `pointmaze_gaint_navigate`     | 39.9 (±5.2) | 43.1 (±1.2) | 72.8 (±2.5) |
> | `pointmaze_giant_stitch`       | 9.1 (±0.8) | 10.2 (±0.9) | 59.2 (±3.2) |
> | `antmaze_large_explore`        | 0.9 (±0.2) | 7.4 (±1.1)  | 67.7 (±2.8)  |
> | `antmaze_giant_stitch`        | 2.7 (±0.6) | 2.4 (±0.4) | 35.1 (±1.7)  |
> | `antmaze_colossal_navigate`          | 22.3 (±1.1) | 38.4 (±1.8) | 48.6 (±2.4) |
> | `humanoidmaze_giant_navigate`      | 9.2 (±1.1)  | 7.4 (±0.5) | 46.5 (±2.5)  |
> | `cube_double_play`      | 13.1 (±2.3)  | 16.2 (±1.4) | 40.8 (±1.2)  |
> | `scene_noisy`      | 19.6 (±1.7)  | 22.1 (±1.7) | 30.8 (±1.6)  |
>
> We see that when we replace learning $C(\pi)$ with multistep backup instead of contrastive learning, there is minor improvement across the board. But since multistep backup does the same job as contrastive learning of recovering the behavior distance, the improvement is not substantial. The main difference in performance can be explained by the fact that while TMD is trying to balance multiple components (learning the behavior distance and optimization), MQE does not do that because it learns a distance by itself.
>
> Unfortunately due to recent events, we cannot push along further discussions, but we did change our manuscript to clearly reflect that difference. We will also fix the incomplete sentence in our camera-ready version. We would also like to thank you again for your constructive comments, which helped the paper to become better.
>
> ---
>
> [1] Myers, V. et al., 2025. "Offline Goal-conditioned Reinforcement Learning with Quasimetric Representations", *NeurIPS*.
>
> [2] Munos, R. et al., 2016. "Safe and Efficient Off-Policy Reinforcement Learning." *NeurIPS*.

---

### Official Review · Reviewer_m7gs · 2025-10-26

**Soundness:** 3
**Presentation:** 2
**Contribution:** 3
**Rating:** 6
**Confidence:** 5

**Summary:**

This paper proposes a new method for offline GCRL. The authors introduce Multistep Quasimetric Estimation (MQE), which applies a multi-step Monte-Carlo return to a quasimetric distance-based method. They evaluate on OGBench and a real robot, where MQE can outperform all baselines.

**Strengths:**

1. The theoretical part is clear and easy to understand. The method is simple, but it still brings a notable improvement in performance. This makes the idea accessible for other researchers and shows that even a straightforward change can lead to meaningful progress.

2. The experiments provide strong comparison results. The authors choose many baselines, including some recent works. This helps confirm that the gains are not due to weak baselines and shows that the method remains strong under fair comparisons.

3. The paper includes real-world testing. The method achieves a higher success rate in practice, and it is able to combine multiple motion segments into complete trajectories. This supports the claim that the approach can work outside simulation.

4. There is enough hyperparameter study. The authors include an analysis that explains how different settings affect performance. This makes the method easier to apply and helps readers understand which parts matter most.

5. The code had been released, showing its reproducibility

**Weaknesses:**

1. While MQE can indeed improve performance, the choice of hyperparameters and the selection of the best waypoint introduce heavy computational cost. This limits the method’s practical use, especially when scaling to more complex tasks or real-time applications.

2. There are many citation errors in the paper, for example, at line 96, 91, 399, and 431. In addition, there are indexing mistakes and incorrect claims. For instance, in Algorithm 1, the loss at line 5 should refer to Eq. 11. Another example is at line 435, where the text claims that TRA-g reaches a positive success rate in the quadruple PnP task, but the table shows 0/15. These issues show that the writing and validation of statements are not strict enough.

3. The value of $\alpha$ used during policy extraction is a hyperparameter that must be tuned separately for different environments. This shows that the method needs heavy hyperparameter tuning to reach the reported performance, which is not practical. I suspect that much of the performance gain shown in the paper comes from tuning these hyperparameters instead of the strength of the method itself.

**Questions:**

1. What happens if the hyperparameters are not tuned, or only tuned with a small number of trials? How much does the performance drop in that case?

2. Can the authors provide theoretical support for using a geometric distribution to select the waypoint? Why is this distribution a reasonable choice compared with other alternatives?

3. In line 386, the paper states that the task "has never been completed without the use of hierarchical policies or high-level planners". Then why is there no comparison against hierarchical policies? Also, can MQE be integrated into a hierarchical framework? If so, why not include such a comparison to better understand its advantages and limitations?

---

> ### Author Response · Authors · 2025-11-21
> **Response**
>
> Thank you for the detailed feedback. It seems like your main concerns are the computational costs of MQE, comparisons to hierarchical methods, and theoretical insights into our waypoint sampling method. To address this, we have **clarified our experimental procedures in this response** and **compared the benefits of our method with other baselines**.
>
> > What happens if the hyperparameters are not tuned, or only tuned with a small number of trials? How much does the performance drop in that case?
>
> We started off with the alpha value reported by CRL, and then chose 4-5 $\alpha$ values that are near the value used by CRL, separated by multiples of 3 (i.e. if CRL used 0.03 as the alpha value, we sweep across $\\{0.003, 0.01, 0.03, 0.1, 0.3\\}$ as the potential hyperparameters). We then used two seeds per alpha value in our hyperparameter tuning procedure, before finding the best alpha value and then running it using 8 random seeds for evaluation. We want to emphasize that this is the same practice used by other offline RL methods [1,2], and we do not incur *additional computational costs*.
>
> This practice (tuning $\\alpha$ per environment over $\\{ 10^{-i}, 3\\cdot 10^{-i} \\}\_{i=0,1,\\dots}$) is used for all of the baselines implemented in the OGBench paper [3] and is in general *required* for offline RL algorithms to work in this setting by imposing conservatism toward $\\pi\_{\\beta}$. Future work should examine eliminating this parameter, but we believe the general problem in offline RL with policy extraction is beyond the scope of this work.
>
> Besides $\\alpha$, the main additional hyperparameter our method needs is the waypoint sampling discount $\\lambda$. The results of ablating $\\lambda$ are shown in Figure 6.
>
> > Can the authors provide theoretical support for using a geometric distribution to select the waypoint? Why is this distribution a reasonable choice compared with other alternatives?
>
> This was an empirical finding. The geometric distribution is appealing for waypoint sampling since its PMF is the same shape as the future state occupancy distribution but with a less heavy tail. Intuitively, this lets us sample waypoints $s^w = s\_{t+k}$ that are able to ''cover'' the full spectrum of future times without sampling past $g=s\_{t+K}$ (i.e., $k\\geq K$) too frequently. We think theoretical analysis of this phenomenon is a good direction for future work, and have included a statement in the limitation section.
>
> > In line 386, the paper states that the task ''has never been completed without the use of hierarchical policies or high-level planners''. Then why is there no comparison against hierarchical policies? Also, can MQE be integrated into a hierarchical framework? If so, why not include such a comparison to better understand its advantages and limitations?
>
> Thank you for bringing this up. We want to note that the hierarchical policies/planners that have successfully completed these tasks are language-conditioned, not image-conditioned. This makes the problem a wholly different one. In general, it is easier to propose subgoals in language-conditioned hierarchical policies as demonstrated in [4,5], and there are no available baselines of an **end-to-end image-based hierarchical policies** on Bridge to our knowledge. While there are policies that use image subgoals during inference [6], it uses a separate large model (Instruct pix2pix, which is based on Stable Diffusion [7]) to produce these images and also requires language interface. We believe that this will be a good direction for future work, and we will modify the text to better scope these claims.
>
> Regarding integrating MQE into a hierarchical framework—an extraction scheme similar to HIQL [8] could be used on top of MQE's learned distance, using a hierarchical policy to output actions. The focus of our work was show how multistep quasimetric learning could extend GCRL to complex settings without additional structure. While hierarchical methods work in OGBench, they require extra complexity to predict subgoals, which complicates the real-world Bridge setup. We believe combining MQE-style distances with hierarchical policy extraction in a scalable fashion is a good direction for future work.

---

> ### Author Response · Authors · 2025-11-21
> **Response (cont.)**
>
> > Another example is at line 435, where the text claims that TRA-g reaches a positive success rate in the quadruple PnP task, but the table shows 0/15.
>
> Thank you for finding this typo. We meant that TRA is the only other method that demonstrated positive success rate in *either* task (GCBC and GCIQL achieved 0 success for both tasks, and TRA achieved positive success rate in `open drawer and place`). We have updated the manuscript to show the difference.
>
> We have also fixed all of the citation errors that you have mentioned, and we would like to thank you for pointing them out. **Do these changes, together with the additional revisions and clarifications discussed below, fully address the reviewer's feedback about the paper?** If not, we look forward to continuing the discussion!
>
> ---
>
> [1] Park, S. et al., 2025. ''Flow Q-Learning.'' *ICML*
>
> [2] Li, Q. et al., 2025. ''Reinforcement Learning With Action Chunking.'' *NeurIPS*
>
> [3] Park, S. et al., 2025. ''OGBench: Benchmarking Offline Goal-Conditioned RL.'' *ICLR*
>
> [4] Myers, V. et al., 2024. ''Policy Adaptation via Language Optimization: Decomposing Tasks for Few-Shot Imitation.'' *CoRL*
>
> [5] Intelligence, P., 2025. ''$\\pi\_{0.5}$: A Vision-Language-Action Model With Open-World Generalization.'' *CoRL*
>
> [6] Black, K. et al., 2024. ''Zero-Shot Robotic Manipulation With Pretrained Image-Editing Diffusion Models.'' *ICLR*
>
> [7] Brooks, T. et al., 2023. ''InstructPix2Pix: Learning to Follow Image Editing Instructions.'' *CVPR*
>
> [8] Park, S. et al., 2023. ''HIQL: Offline Goal-Conditioned RL With Latent States as Actions.'' *NeurIPS*

---

> > ### Comment · Reviewer_m7gs · 2025-11-24
> >
> > Thank you for the detailed responses. Regarding Answer 3, I would like to see the experiment that integrates MQE into HIQL. I believe it will make the paper more solid.

---

> ### Author Response · Authors · 2025-11-27
> **MQE+HIQL Policy Extraction**
>
> We have provided the results to preliminary experiments of MQE using the HIQL-style policy extraction method below:
>
> | Dataset                              | HIQL        | MQE + HIQL policy extraction      | MQE          |
> | ------------------------------------ | ------------ | --------------------- | ------------ |
> | `pointmaze_gaint_navigate`           | 45.9 (±3.0)   | 69.1 (±2.5)            | 72.8 (±2.5)  |
> | `pointmaze_giant_stitch`             | 0.0 (±0.0)   | 67.3 (±4.8)            | 59.2 (±3.2)  |
> | `antmaze_giant_stitch`               | 1.8 (±0.6)   | 21.2 (±1.7)            | 26.5 (±1.3)  |
> | `humanoidmaze_giant_navigate`        | 12.5 (±1.5)   | 77.4 (±1.9)            | 46.5 (±1.3)  |
> | `cube_double_play`                   | 6.4 (±0.7)  | 44.7 (±1.3)            | 40.8 (±1.2)  |
>
> Generally, we find that incorporating HIQL policy extraction tends to improve the overall performance on some of the most difficult tasks. One important caveat is that HIQL only used AWR-style policy extraction, while previous work [1] has shown that DDPG+BC policy extraction (which we have used for MQE) might be better for improving performance. To compare against HIQL fairly, we used AWR for policy extraction in our MQE+HIQL implementation, but we can certainly explore the alternative choice of incorporating DDPG+BC in the future to see if there are additional gains.
>
> We will conduct further experiments and explore alternative policy extraction design choices in the following days across other environments and datasets, and we will incorporate other ways of policy extraction in the camera-ready version to demonstrate the effects of such design choices.
>
> **Are there any additional experiments, clarifications, or other information we could provide?** We believe we have revised and polished the manuscript to address your concerns, and are happy to share more details or make further revisions if you have additional reservations.
>
> ---
>
> [1] Park, S et al., 2024. “Is Value Learning Really the Main Bottleneck in Offline RL?”, *NeurIPS*

---

### Official Review · Reviewer_4Ky7 · 2025-10-28

**Soundness:** 4
**Presentation:** 4
**Contribution:** 4
**Rating:** 8
**Confidence:** 3

**Summary:**

The paper introduces Multistep Quasimetric Estimation (MQE), a novel goal-conditioned reinforcement learning (GCRL) method that unifies temporal-difference (TD) and Monte Carlo (MC) learning through a quasimetric distance representation. MQE leverages multistep returns under a quasimetric architecture to propagate value information efficiently across long horizons while maintaining theoretical consistency with optimal value functions. It further enforces action invariance and one-step consistency constraints to stabilize learning. Empirically, MQE achieves state-of-the-art performance on long-horizon offline GCRL benchmarks (up to 4000 steps) and demonstrates strong compositional generalization in real-world robotic manipulation tasks. Its key contribution is showing that multistep temporal consistency and quasimetric structures can be combined to enable scalable, end-to-end goal-reaching from raw, unlabeled offline data.

**Strengths:**

This paper makes an original contribution by unifying multistep value learning and quasimetric representations into a single, scalable framework for goal-conditioned reinforcement learning. The idea of integrating multistep temporal consistency with quasimetric architectures is both novel and technically elegant, addressing long-standing limitations in horizon generalization and stability. The paper is of high quality, with strong theoretical grounding, clear algorithmic exposition, and comprehensive experiments across both simulated and real-world robotic domains. Its clarity allows readers to follow complex ideas with precision, and its significance lies in demonstrating a practical path toward scalable, compositional goal-reaching in offline RL—bridging a crucial gap between theory and real-world applicability.

**Weaknesses:**

While the paper is strong overall, several aspects could be improved to strengthen its impact. The theoretical analysis, while elegant, could benefit from clearer discussion of its assumptions and limitations—particularly regarding stability and convergence in high-dimensional or stochastic environments. Finally, the presentation could be improved by providing more intuition and visualization of the learned quasimetric distances to help readers better grasp the geometric and representational properties driving the observed performance gains.

**Questions:**

1.Could the authors clarify the conditions under which the proposed multistep quasimetric estimation guarantees convergence or consistency? Specifically, how sensitive are these guarantees to function approximation errors or off-policy data distributions?

2.The method integrates multistep temporal consistency into the quasimetric framework, but it remains unclear how much of the empirical gain arises from longer-horizon updates versus the quasimetric structure itself. Could the authors provide quantitative or qualitative evidence disentangling these effects (e.g., via controlled ablations or visualizations)?

---

> ### Author Response · Authors · 2025-11-21
> **Response**
>
> Thank you for your detailed feedback. It seems like your main concerns relate to presentation, theoretical justification of MQE, and comparing against additional baselines that use n-step returns. **We have added [theoretical analysis](https://res.cloudinary.com/dp7qzzmt2/image/upload/v1763727124/MQE_Rebuttal-1763726716266_c8fjq1.png)** of the MQE objective, and **added two additional n-step baselines**. We hope these revisions fully address your concerns.
>
> > Could the authors clarify the conditions under which the proposed multistep quasimetric estimation guarantees convergence or consistency? Specifically, how sensitive are these guarantees to function approximation errors or off-policy data distributions?
>
> Thank you for bringing this up. We have provided a theoretical justification in the main text [(Theorem 1)](https://res.cloudinary.com/dp7qzzmt2/image/upload/v1763727124/MQE_Rebuttal-1763726716266_c8fjq1.png) to demonstrate the convergence of the MQE objective. In short, MQE allows policy improvement upon the behavior policy $\pi\_\\beta$, which is the reason behind significant gains over methods such as CMD.
>
> > The method integrates multistep temporal consistency into the quasimetric framework, but it remains unclear how much of the empirical gain arises from longer-horizon updates versus the quasimetric structure itself. Could the authors provide quantitative or qualitative evidence disentangling these effects (e.g., via controlled ablations or visualizations)?
>
> Thank you for bringing this up. We compared against QRL (which does local value updates) to see the importance of multistep value returns, and we have also implemented multistep GCIQL to see whether the same multistep returns can be beneficial to the same degree to non-quasimetric methods. We present the results below:
>
>
> | Dataset                          | GCIQL       | Multistep GCIQL       | MQE | |
> | ------------------------------------ | ------------ | --------------------- | ------------ | --- |
> | `pointmaze_gaint_navigate`     | 0.0 (±0.0) | 4.1 (±1.1) | 72.8 (±2.5) | |
> | `pointmaze_giant_stitch`       | 0.0 (±0.0) | 0.0 (±0.0) | 59.2 (±3.2) | |
> | `antmaze_large_explore`        | 0.4 (±0.1) | 2.2 (±0.7)  | 67.7 (±2.8)  |   |
> | `antmaze_giant_stitch`        | 0.0 (±0.0) | 0.0 (±0.0) | 26.5 (±1.3)  | |
> | `antmaze_colossal_navigate`          | 0.0 (±0.0) | 5.4 (±1.3) | 48.6 (±2.4) |   |
> | `humanoidmaze_giant_navigate`      | 0.5 (±0.1)  | 1.1 (±0.3)  | 46.5 (±1.3)  |   |
> | `cube_double_play`      | 40.2 (±1.7)  | 5.4 (±1.5)  | 40.8 (±1.2)  | |
> | `scene_noisy`      | 25.9 (±0.8)  | 13.2 (±1.4)  | 30.8 (±1.4)  | |
>
>
> A few experiments have not yet finished, hence the missing entries. We will continue to update the table throughout the rebuttal process, although the experiments conducted so far point to the conclusion that a multistep loss on GCIQL does not improve the performance of the policy.
>
> **Do these changes, together with the additional revisions and clarifications discussed below, fully address the reviewer's feedback about the paper?** If not, we look forward to continuing the discussion!

---

> > ### Comment · Reviewer_4Ky7 · 2025-11-27
> >
> > **We appreciate the authors’ response. I have read it carefully and decided to keep my original rating.**

---

### Official Review · Reviewer_pzjw · 2025-10-29

**Soundness:** 3
**Presentation:** 2
**Contribution:** 4
**Rating:** 6
**Confidence:** 3

**Summary:**

This paper addresses the challenge of scaling RL to real-world, long-horizon robotics-tasks via offline GCRL. The main contribution of the paper is a novel algorithm, Multistep Quasimetric Estimation (MQE). The main technical innovation is how to learn multistep returns under quasimetric architectures while preserving the ability to learn the optimal value function. The authors show that MQE obtains state-of-the-art results in challenging long-horizon GCRL tasks in OGBench and a real-world robotics task.

**Strengths:**

- The paper addresses an fundamental challenge in RL - namely, how can we scale to real-world tasks? Personally, I think the approach of trying to induce compositional learning via offline GCRL, succ, and goal-stitching is very promising. The topic will definitely be interesting to the ICLR community.

- The proposed approach is empirically highly effective compared to baselines in state-based datasets and the real-world robotics task. Overall, the evaluation is thorough. There is a large-scale evaluation of many methods and tasks.

**Weaknesses:**

- **Writing:** the writing is the biggest weakness of this paper. Unfortunately, the authors focus on explaining technical details of the method to an audience of experts in offline RL / GCRL, while failing to communicate high level, key messages to a general audience (see below). I’m concerned that this will limit its interest/impact to the general ICLR audience.

    - The motivation and the problem being solved is not very coherent at the beginning of the paper. I found myself wondering for a while whether the authors were addressing GCRL, or offline RL. It took until I almost finished skim-reading the paper to confirm that the authors address offline GCRL. Looking back at the paper, I think this occurred because the abstract only addresses GCRL rather than offline GCRL, and the term “Offline GCRL” only appears in conjunction in 2 places in the manuscript.

    - What is the key insight of the paper? Why do multistep returns matter so much torwards improving performance on the long-horizon offline tasks?

    - Some minor typos in the citations:

        - Please make sure to use citep where appropriate.

        - See Lines 91, 96, 122, 399

    - Lack of clarity:

        - What is task prowess metric in Fig 3 and how does it differ from success rate?

- **Soundness of Method**: The proposed method relies on trick of sampling next states w/ higher prob according to a Bernoulli distribution, thus indicating the auxiliary loss in 4.2 is not sufficient to fix bias issues. Can the authors comment on this point? What is the relative contribution of this trick to the final performance of MQE vs the auxiliary loss?

**Questions:**

- The waypoints are randomly sampled according to the geometric distribution. Are there any theoretical benefits of geom distr. over other distributions, esp those considered in App. E?

- What is the benefit of multi-step return in this setting? From another angle, what is the problem with single step returns?

- Intuitively, why does Lp violate optimality while Lt doesn't? Can the authors provide an illustrative example or proof?

- Expts:

    - Why does MQE do particularly poorly on cube_triple_noisy?

    - Why does MQE underperform baselines on visual tasks, especially compared to HIQL?

    - Why not compare against HIRL in Bridge Data?

- Is this method applicable to online settings?

---

> ### Author Response · Authors · 2025-11-21
> **Response**
>
> Thank you for the thoughtful feedback. It seems your main concerns relate to writing and presentation, as well as providing justification on the convergence of the method. We have **substantially revised the text based on your and other reviewers' feedback**, with key changes highlighted in orange, as well as **[including theory](https://res.cloudinary.com/dp7qzzmt2/image/upload/v1763727124/MQE_Rebuttal-1763726716266_c8fjq1.png) that details how MQE performs policy improvement beyond the behavioral Q function $Q\_\\beta$**.
>
> > What is the benefit of multi-step return in this setting? From another angle, what is the problem with single step returns?
>
> There is an inherent tradeoff between multistep returns and single-step returns, where single-step returns are enforcing (theoretically optimal) local consistency while multistep returns are encouraging faster global value propagation, at the cost of optimality [1,2]. Empirically, we find out that using single step returns does not help learn a good value (thus extract a good policy) representation over the quasimetric, which is reflected in **Table 1b of our revised manuscript** in the experiment section, as it achieved 0% success rate in our ablation.
>
> > The waypoints are randomly sampled according to the geometric distribution. Are there any theoretical benefits of geom distr. over other distributions, esp those considered in App. E?
>
> This was an empirical finding. The geometric distribution is appealing for waypoint sampling since its PMF is the same shape as the future state occupancy distribution but with a less heavy tail. Intuitively, this lets us sample waypoints $s^w = s\_{t+k}$ that are able to ''cover'' the full spectrum of future times without sampling past $g=s\_{t+K}$ (i.e., $k\\geq K$) too frequently. We think theoretical analysis of this phenomenon is a good direction for future work, and have included a statement in the limitation section.
>
> > The proposed method relies on trick of sampling next states w/ higher prob according to a Bernoulli distribution, thus indicating the auxiliary loss in 4.2 is not sufficient to fix bias issues. Can the authors comment on this point? What is the relative contribution of this trick to the final performance of MQE vs the auxiliary loss?
>
> We assume that the auxiliary loss here refers to the action invariance objective. We would like to clarify that the action invariance objective regresses the value function to the maximum of the Q value [3], which corresponds to greedy value estimation. In the non-quasimetric setting, this is similar to IQL's value loss with $\\tau \\approx 1$ [4].
>
> Therefore, the sampling trick is less related to the idea of action invariance, but more related to the idea of manually up-weighting the single-step $\\mathcal{T}$ backup and how to combine it with the more biased behavioral backup. Empirically, we find that using a Bernoulli + Geometric distribution works the best, and we have recorded each sampling decision in table 1 in the main manuscript. We have modified our manuscript to clearly delineate such a difference
>
> > Intuitively, why does Lp violate optimality while Lt doesn't? Can the authors provide an illustrative example or proof?
>
> The reasoning behind this statement is that the multistep loss $\\mathcal{L}\_{\\mathcal{T}\_\\beta}$ is essentially performing a multistep backup under a quasimetric architecture, whereas $\\mathcal{L}\_{\\mathcal{T}}$ is only performing single-step backup. As a result, the same reasoning for suboptimality can be applied from previous works on multistep off-policy Q learning [5,6]. Intuitively, multistep returns will be suboptimal because the future state is being conditioned on the actions taken by the behavior policy, and thus $\\mathcal{T}\_{\\beta}$ incorporates information about both the dynamics and $\\pi\_{\\beta}$, unlike $\\mathcal{T}$ which just operates on the dynamics.
>
> > Why does MQE do particularly poorly on cube\_triple\_noisy?
>
> Thank you for bringing this up. We made adjustments in the action invariance objective in section 4 and updated the results (see revised Table 4 in Appendix B), in which cube\_triple\_noisy improved its performance considerably.
>
> > Why does MQE underperform baselines on visual tasks, especially compared to HIQL?
>
> We would like to emphasize that MQE outperforms all baselines on visual tasks except HIQL. Regarding HIQL, one important distinction is that HIQL uses a high and low level policy, which doubles the amount of learnable parameters [7]. In addition, HIQL's low level policy only takes in one visual observation (and takes in an already-encoded goal), while other policies take in both visual observations concurrently. We believe this explains HIQL's strong performance on pixel based tasks, where such clear hierarchy is not shown in other methods.

---

> > ### Author Response · Authors · 2025-11-21
> > **Response (cont.)**
> >
> > > Why not compare against HIRL in Bridge Data?
> >
> > We assume you are referring to Hierarchical Inverse Reinforcement Learning [8]. In practice, hierarchical methods in Bridge require additional structure, such as LLM planners [9] or generative image models [10] to perform subtask selection. As such, we focused the real-world evaluation on end-to-end non-hierarchical GCRL methods for fair comparison. We compare against the hierarchical HIQL method in our simulated OGBench experiments, where HIQL is able to perform well without additional structure.
> >
> > > What is task progress metric in Fig 3 and how does it differ from success rate?
> >
> > In Figure 3, we defined task progress as how many consecutive tasks did the policy successfully execute, divided by the total number of tasks present. For example, if a policy completes one out of four requisite pick and place tasks in `quadruple pnp`, then it would receive a score of 0.25, and in the case of single-stage tasks, the task progress metric is equivalent to success rate. We include task success rates in the appendix as well, but task progress is useful since it allows more fine-grained comparison of methods.
> >
> > **Do these changes, together with the additional revisions and clarifications discussed below, fully address the reviewer's feedback about the paper?** If not, we look forward to continuing the discussion!
> >
> > ---
> >
> > [1] Park, S. et al., 2025. ''Horizon Reduction Makes RL Scalable.'' *NeurIPS*
> >
> > [2] Hernandez-Garcia, J. F., 2019. ''Understanding Multi-Step Deep Reinforcement Learning: A Systematic Study of the DQN Target.'' arXiv:1901.07510
> >
> > [3] Myers, V. et al., 2024. ''Learning Temporal Distances: Contrastive Successor Features Can Provide a Metric Structure for Decision-Making.'' *ICML*
> >
> > [4] Kostrikov, I. et al., 2022. ''Offline Reinforcement Learning With Implicit Q Learning.'' *ICLR*
> >
> > [5] Kozuno, T. et al., 2021. ''Revisiting Peng's Q($\\lambda$) for Modern Reinforcement Learning.'' *ICML*
> >
> > [6] Munos, R. et al., 2016. ''Safe and Efficient Off-Policy Reinforcement Learning.'' *NeurIPS*
> >
> > [7] Park, S. et al., 2023. ''HIQL: Offline Goal-Conditioned RL With Latent States as Actions.'' *NeurIPS*
> >
> > [8] Krishnan, S. et al., 2016. ''HIRL: Hierarchical Inverse Reinforcement Learning for Long-Horizon Tasks With Delayed Rewards.'' *ArXiv*
> >
> > [9] Michał, Z. et al., 2024. ''Robotic Control via Embodied Chain-of-Thought Reasoning.'' arXiv:2407.08693
> >
> > [10] Black, K. et al., 2024. ''Zero-Shot Robotic Manipulation With Pre-Trained Image-Editing Diffusion Models.'' *ICLR*

---

> > ### Comment · Reviewer_pzjw · 2025-11-27
> >
> > Thank you for the revisions to the manuscript and your detailed replies to my comments. My concerns have been addressed. I will maintain my score, which reflects my belief that the paper should be accepted.

---

### Official Review · Reviewer_sYkc · 2025-10-31

**Soundness:** 2
**Presentation:** 1
**Contribution:** 3
**Rating:** 4
**Confidence:** 3

**Summary:**

This paper introduces Multistep Quasimetric Estimation (MQE), an offline GCRL method that integrates multistep returns within a quasimetric network architecture.
The paper aims to reconcile the tension between the theoretical optimality of local TD updates and the superior horizon scaling of global Monte-Carlo methods.
MQE updates values by regressing distances using geometrically sampled "waypoints" and enforcing action invariance.
MQE is evaluated on complex stitching and noisy tasks from OGBench where it outperforms common baselines, including often outperforming the hierarchical HIQL.
Moreover, the paper presents impressive compositional generalization in real-world robotic manipulation.

**Strengths:**

The paper's main strength is the integration of multistep backups into the quasimetric learning framework.
This enables superior horizon generalization compared to prior methods, which the paper demonstrates extensively on challenging environments from OGBench requiring extremely long-horizon planning.
The success in the real-world BridgeData experiments is compelling.
To my knowledge, complex multi-stage compositionality (e.g., Quadruple Pick and Place) using a flat (non-hierarchical) policy architecture represents significant progress in scalable GCRL.

**Weaknesses:**

* The multistep backup (Eq. 9) is inherently biased towards the behavior policy​ when the step k′>1. While the authors mitigate this with 1-step consistency weighting and action invariance, the justification for how this combination robustly overcomes the strong bias of the multistep return is primarily empirical.
* In my view, the paper is missing a comparison to recent work [1] that addresses the identical challenge of achieving long-horizon GCRL performance with a flat policy.

The presentation has several issues:
* The related work section has several missing or ill-formatted citations.
* The paper has several incomplete sentences, e.g., line 91.
* Page 5 includes a lot of extra spacing.
* Overall, I find the methods section (Section 4) hard to follow. It would be beneficial to add structure to the section and provide additional guidance to the reader.

**Questions:**

* Is the quasimetric architecture essential for the success of the multistep backup, or could this backup strategy also improve standard value-based methods (e.g., IQL) without the explicit distance structure?

Note: The paper title of the PDF does not match the title on OpenReview.

---

> ### Author Response · Authors · 2025-11-21
> **Response**
>
> Thank you for your detailed feedback. It seems like the main concerns relate to presentation, theoretical justification of MQE, and comparing against additional baselines that use n-step returns. **We have revised the paper (changes highlighted in orange)** based on your suggestions, **added theoretical analysis** of the MQE objective, and **added two additional n-step baselines**. We hope these revisions will address your concerns.
>
> > In my view, the paper is missing a comparison to recent work \[1] that addresses the identical challenge of achieving long-horizon GCRL performance with a flat policy.
>
> Could you clarify which work \[1] refers to?
>
> > The multistep backup (Eq. 9) is inherently biased towards the behavior policy​ when the step k′>1. While the authors mitigate this with 1-step consistency weighting and action invariance, the justification for how this combination robustly overcomes the strong bias of the multistep return is primarily empirical.
>
> Thank you for your feedback. We have included theoretical results [(Theorem. 1)](https://res.cloudinary.com/dp7qzzmt2/image/upload/v1763727124/MQE_Rebuttal-1763726716266_c8fjq1.png) in our manuscript to justify convergence of MQE. In short, MQE can be shown to perform policy improvement on top of $\pi_\beta$ under standard assumptions.
>
> > The presentation has several issues:...
>
> We have made revisions to our manuscript to address your concerns. We have resolved the issue of spacing, incomplete sentences, and citations that you have brought up. We have revised our manuscript to provide more clarity, and we have corrected all of the typos.
>
> >Is the quasimetric architecture essential for the success of the multistep backup, or could this backup strategy also improve standard value-based methods (e.g., IQL) without the explicit distance structure?
>
> Thank you for bringing this up. When the same multistep backup strategy is applied on non-quasimetric methods, we discovered that it produces little to no benefits in horizon generalization. We implemented multistep GCIQL using the the same future goal sampling tactic with GCIQL and waypoint sampling tactics, and we presents the results below:
>
> | Dataset                              | GCIQL        | Multistep GCIQL       | MQE          |     |
> | ------------------------------------ | ------------ | --------------------- | ------------ | --- |
> | `pointmaze_gaint_navigate`           | 0.0 (±0.0)   | 4.1 (±1.1)            | 72.8 (±2.5)  |     |
> | `pointmaze_giant_stitch`             | 0.0 (±0.0)   | 0.0 (±0.0)            | 59.2 (±3.2)  |     |
> | `antmaze_large_explore`              | 0.4 (±0.1)   | 2.2 (±0.7)            | 67.7 (±2.8)  |     |
> | `antmaze_giant_stitch`               | 0.0 (±0.0)   | 0.0 (±0.0)            | 26.5 (±1.3)  |     |
> | `antmaze_colossal_navigate`          | 0.0 (±0.0)   | 5.4 (±1.3)            | 48.6 (±2.4)  |     |
> | `humanoidmaze_giant_navigate`        | 0.5 (±0.1)   | 1.1 (±0.3)            | 46.5 (±1.3)  |     |
> | `cube_double_play`                   | 40.2 (±1.7)  | 5.4 (±1.5)            | 40.8 (±1.2)  |     |
> | `scene_noisy`                        | 25.9 (±0.8)  | 13.2 (±1.4)           | 30.8 (±1.4)  |     |
>
> A few experiments have not yet finished, hence the missing entries. We will continue to update the table throughout the rebuttal process, although the experiments conducted so far point to the conclusion that a multistep loss on GCIQL does not improve the performance of the policy.
>
> > Title of the paper is different from OpenReview title
>
> The current title of the manuscript is “Multistep Quasimetric Learning for Scalable Goal-conditioned Reinforcement Learning.” We will be able to correct the OpenReview title for the camera ready submission.
>
> **Do these changes fully address your concerns?** We look forward to continuing the discussion!

---

> > ### Comment · Reviewer_sYkc · 2025-11-24
> >
> > Thank you for your detailed responses. Apologies for the missing reference in my original review:
> >
> > [1] Zhou & Kao. Flattening Hierarchies with Policy Bootstrapping. In NeurIPS 2025.
> >
> > Can you clarify how your work relates to this method which addresses the same challenge of solving long-horizon tasks with flat GCRL policies? In my view an empirical comparison on OGBench would be valuable as well.

---

> > > ### Author Response · Authors · 2025-11-27
> > > **Comparisons**
> > >
> > > Thank you for bringing this paper up. From what we have seen, the method presented in the paper (which we will refer to as SAW) you have mentioned proposed an alternative version of policy extraction, as SAW aligns the output of the policy trained on long-horizon goals to the policy trained on short-horizon goals. Therefore, the way that SAW addresses solving long-horizon tasks with GCRL is actually orthogonal to what we have proposed. The key value learning aspect of SAW is the exact same as GCIVL (and by extension, HIQL), and as a result, we can replace it with MQE's critic/value learning objectives.
> > >
> > > We have provided comparisons between SAW, MQE, and SAW+MQE in the table below. SAW+MQE uses MQE to learn the critic and value parameterization, and does policy extraction using the SAW objective, which uses AWR throughout the entire algorithm.
> > >
> > > | Dataset                              | SAW        | SAW+MQE       | MQE          |
> > > | ------------------------------------ | ------------ | --------------------- | ------------ |
> > > | `pointmaze_gaint_navigate`           | 68 (±8)   | 74.1 (±1.1)            | 72.8 (±2.5)  |
> > > | `humanoidmaze_giant_navigate`        | 35 (±5)   | 63.9 (±0.8)            | 46.5 (±1.3)  |
> > > | `scene_play`        | 63 (±6)   | 73.1 (±1.9)            | 76.8 (±2.1)  |
> > > | `cube_double_play`                   | 40 (±7)  | 45.4 (±2.2)            | 40.8 (±1.2)  |
> > >
> > > In the experiments against SAW, we have also used AWR for SAW+MQE's policy extraction method to maintain a fair comparison. Across the board, the SAW objective seems to marginally help policy extraction compared to vanilla DDPG+BC. However, we remark that previous works [1] have shown that DDPG+BC tend to outperform AWR-based policy extraction methods in off-policy RL.
> > >
> > > We will conduct further experiments and explore alternative policy extraction design choices in the following days across other environments and datasets. We will also incorporate other ways of policy extraction in the camera-ready version to demonstrate the effects of such design choices.
> > >
> > > **Are there any additional experiments, clarifications, or other information we could provide?** We believe we have revised and polished the manuscript to address your concerns, and are happy to share more details or make further revisions if you have additional reservations.
> > >
> > > ---
> > >
> > > [1] Park, S et al., 2024. “Is Value Learning Really the Main Bottleneck in Offline RL?”, *NeurIPS*

---

> > > > ### Comment · Reviewer_sYkc · 2025-11-27
> > > >
> > > > Thank you for your detailed response. I believe that your revisions during the discussion phase have substantially improved the paper, both in terms of readability and presentation as well as in terms of breadth of experiments.
> > > > Reflacting this, I have raised my score to recommend acceptance.

---

### Author Response · Authors · 2025-12-03
**Summary of Discussions**

Dear Reviewers and ACs,

We would like to thank you for participating in the discussion period. We acknowledge that unprecedented events have prevented us from having further discussions, nevertheless, the discussion period has been productive and helped to improve the paper. We present a recap on the concerns of each reviewer, and how we addressed them to improve the paper. We want to note that reviewer sYkc have **raised their score from 4 to 8** and advocated for the paper’s acceptance, and reviewer jD1e have **raised their score from 2 to 4 but did not finish the disucssion** prior to the scores being reverted. Other reviewers have maintained their score, and advocated for the paper’s acceptance as well.

> Theoretical Justification (Reviewer sYkc, pzjw, 4Ky7)

We showed that MQE does policy improvement beyond $\pi_\beta$ in our paper’s appendix section [(Theorem. 1)](https://res.cloudinary.com/dp7qzzmt2/image/upload/v1763727124/MQE_Rebuttal-1763726716266_c8fjq1.png).

> Comparison against multistep non-quasimetric methods (Reviewers sYkc, 4Ky7)

We have implemented multistep GCIQL, and we have shown that multistep GCIQL does not achieve performance gains compared to GCIQL in our discussions.

> Comparison against other policy extraction methods (Reviewers sYkc, m7gs)

We have compared against SAW [1] and have implemented MQE using SAW-style policy extraction, and show considerable performance gains against SAW and some gains against MQE using ddpg+bc. We have also shown that when using HIQL-style policy extraction, it improves the performance of the policy as well as HIQL explicitly reduces the policy horizon.

> Novelty of the paper (Reviewer jD1e)

During our rebuttals, we explained that conceptually and in practice, TMD and MQE are considerably different algorithms, and that MQE’s main contribution was a scalable quasimetric reinforcement learning method that allows real-world success. We have also shown the difference between MQE and TMD in our manuscript.

> Improvements on Presentation (Reviewer sYkc, pzjw, 4Ky7, jD1e)

We have significantly modified the paper to incorporate the feedback of the authors in terms of presentation. All of the changes are highlighted in beige in the revised manuscript.

> Discussion on computational costs and waypoint sampling methods (Reviewer m7gs, pzjw)

We have shown that compared to prior works in offline RL, MQE incurs little additional computational cost in terms of hyperparameter search. We also note that the waypoint sampling method is empirical, and we have stated theoretical justification of waypoint sampling can be explored in future work in the paper.

We have also updated our experimental evaluations in the paper, which demonstrated superior performance of MQE against other methods.

---

[1] Zhou, J et al., 2025. “Flattening Hierarchies with Policy Bootstrapping”. *NeurIPS*.

---

### Meta-Review · Area_Chair_JpE8 · 2025-12-25

**Summary:**

The reviewers generally found the paper technically sound and empirically promising, with strengths in methodology and experimental results. However, concerns were raised regarding the novelty relative to closely related work, the completeness and rigor of the evaluation, and aspects of clarity and justification. The rebuttal addressed several technical and experimental issues through clarifications and additional explanations, but one reviewer remains unconvinced about the level of conceptual novelty.

**Reviewer Concerns:**

**Addressed by the rebuttal:**

The authors clarified the method’s relationship to prior work, explaining key differences and intended contributions.

Additional explanations improved the theoretical motivation and algorithmic design, addressing confusion raised by multiple reviewers.

The rebuttal strengthened the experimental discussion, helping justify performance gains and design choices.

**Still outstanding:**

Reviewer jD1e remains concerned that the method represents an incremental extension rather than a clearly novel contribution.

Some reviewers noted that evaluation depth (e.g., ablations, broader validation) could still be improved, even if partially addressed in the rebuttal.

**Reviewer Scores:**

Reviewer sYkc: Likely more positive after the rebuttal.

Reviewer pzjw: Likely unchanged, remaining supportive of acceptance.

Reviewer 4Ky7: Likely unchanged, maintaining a positive evaluation.

Reviewer jD1e: Slightly softened but borderline.

---

### Decision · Program_Chairs · 2026-01-26

Accept (Poster)